# Towards World Simulator:
# Crafting Physical Commonsense-Based Benchmark for Video Generation

**Fanqing Meng** [* 1 2]  **Jiaqi Liao** [* 2]  **Xinyu Tan** [2]  **Quanfeng Lu** [1 2]  **Wenqi Shao** [2]  **Kaipeng Zhang** [2]  **Yu Cheng** [3]
**Dianqi Li**  **Ping Luo** [2 4]

## Abstract

Text-to-video (T2V) models like Sora have made significant strides in visualizing complex prompts, which is increasingly viewed as a promising path towards constructing the universal world simulator. Cognitive psychologists believe that the foundation for achieving this goal is the ability to understand intuitive physics. However, the capacity of these models to accurately represent intuitive physics remains largely unexplored. To bridge this gap, we introduce *PhyGenBench*, a comprehensive **Phy**sics **Gen**eration **Bench**mark designed to evaluate physical commonsense correctness in T2V generation. *PhyGenBench* comprises 160 carefully crafted prompts across 27 distinct physical laws, spanning four fundamental domains, which could comprehensively assess models' understanding of physical commonsense. Alongside *PhyGenBench*, we propose a novel evaluation framework called *PhyGenEval*. This framework employs a hierarchical evaluation structure utilizing appropriate advanced vision-language models and large language models to assess physical commonsense. Through *PhyGenBench* and *PhyGenEval*, we can conduct large-scale automated assessments of T2V models' understanding of physical commonsense, which aligns closely with human feedback. Our evaluation results and in-depth analysis demonstrate that current models struggle to generate videos that comply with physical commonsense. Moreover, simply scaling up models or employing prompt engineering techniques is insufficient to fully address the challenges presented by *PhyGenBench* (e.g., dynamic physical phenomenons). We hope this study will inspire the community to prioritize the learning of physical commonsense in these models beyond entertainment applications. We release the data and codes at `https://github.com/OpenGVLab/PhyGenBench`

---

[*]Equal contribution  [1]Shanghai Jiao Tong University  [2]Shanghai AI Laboratory  [3]The Chinese University of Hong Kong  [4]The University of Hong Kong.  Correspondence to: Wenqi Shao <shaowenqi@pjlab.org.cn>, Ping Luo <pluo@cs.hku.hk>.

*Proceedings of the 42nd International Conference on Machine Learning*, Vancouver, Canada. PMLR 267, 2025. Copyright 2025 by the author(s).

## 1. Introduction

Text-to-video (T2V) models such as Sora have made significant strides in visualizing complex ideas and scenes based on textual input (Yang et al., 2024; Wang et al., 2023). These advancements are increasingly viewed as a promising path towards constructing universal simulators of the physical world, which holds immense promise for video generation (Zhu et al., 2024), autonomous driving (Gao et al., 2024), and the development of embodied agents (Mazzaglia et al., 2024). Cognitive psychology posits that intuitive physics, which is demonstrated even by human infants (Wood et al., 2024; Battaglia et al., 2013), is essential for achieving this goal. Intuitive physics emphasizes rendered scenes should be visually and interactively natural to humans, rather than adhere to strict physical accuracy. Consequently, on the path towards developing a world simulator (Xiang et al., 2024), video generation should first be capable of accurately reproducing simple yet fundamental physical phenomenons. However, even state-of-the-art models trained on vast resources (Tan et al., 2024) encounter difficulties in correctly generating seemingly trivial physical phenomenons, as depicted in Figure 1, the model fails to understand that the stone should sink in water. This clear pitfall shows a substantial gap between current video generation models' and human's understanding of basic physics. It reveals how far these models are from being true world simulators.

Given this context, it becomes important to assess the extent to which current T2V models can capture intuitive physics in their generated outputs. This requires the development of comprehensive evaluation frameworks that beyond traditional metrics. While numerous Text-to-Video (T2V) evaluation benchmarks have emerged (Sun et al., 2024; Huang et al., 2024), they primarily focus on various qualities of generated videos (e.g., motion smoothness, background

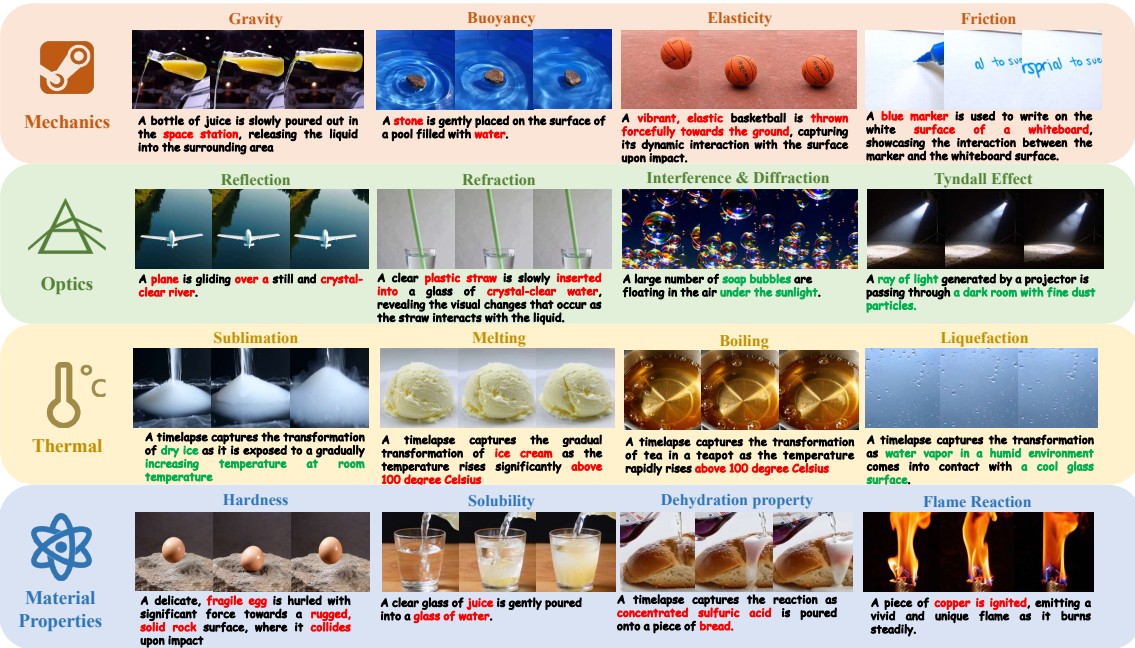

Figure 1: Samples of videos generated by Kling or Gen-3 in *PhyGenBench* with 4 different aspects. The results show that current T2V models struggle to generate videos that align with physical commonsense (e.g., the lack of a plane's reflection in water in the first video of the second row).

consistency) or spatial relationships, failing to address the critical issue of whether the generated videos adhere to fundamental physical laws. Although some studies have explored the alignment of generated videos with dynamic motions naturalness (Bansal et al., 2024), their benchmarks fail to succinctly capture fundamental physical laws or propose sufficiently robust evaluation methods. Therefore, the development of benchmarks and evaluation methodologies specifically tailored to assess intuitive physics in generated videos remains a critical yet largely unexplored frontier.

There are two challenges impeding the evaluation of physical commonsense in T2V models. On one hand, there is a lack of benchmarks focused on evaluating physical commonsense. This requires selecting semantically simple physical phenomenons that exhibit clear physical phenomena, allowing for accurate assessment by either humans or machines. On the other hand, there is a lack of corresponding evaluation metrics. Traditional metrics like FVD (Unterthiner et al., 2018) exhibit limitations in detecting implausible motions (Brooks et al., 2022) and necessitate reference videos, which are often challenging to procure for novel scenes. Recent studies have used video-based VLMs for comprehensive video evaluation (He et al., 2024b; Sun et al., 2024). However, they often struggle to correctly assess physical commonsense. This limitation stems from their inadequate understanding of physical laws (Jassim et al., 2023) and the fact that these methods are not specifically designed to evaluate physical laws.

To address these challenges, we propose *PhyGenBench* and *PhyGenEval* to automate the evaluation of physical commonsense understanding capability from T2V models. *PhyGenBench* is designed to evaluate physical commonsense based on fundamental physical laws in text-to-video generation. Inspired by (Halliday et al., 2013), we categorize physical commonsense in the world into four main areas: mechanics, optics, thermal, and material properties. Then principle physical laws and easily observable physical phenomenons are identified for each category, resulting in comprehensive 27 physical laws and 160 validated prompts in the proposed benchmark. Through brainstorming, we construct prompts that easily reflect physical laws using sources like textbooks (Harjono et al., 2020). This process results in a comprehensive but simple set of prompts reflecting physical commonsense, which are sufficiently clear for evaluation. As shown in Figure 1, the correctness of physical commonsense in *PhyGenBench* can be observed through clear phenomena (e.g., *plane should have reflections in water*) On the other hand, benefiting from the simple yet clear physical phenomena in *PhyGenBench* prompts, we can propose *PhyGenEval*, which is a novel video evaluation framework for assessing physical commonsense correctness in *PhyGenBench*. *PhyGenEval* first uses GPT-4o[1] to analyze physical laws in text, addressing the poor understanding of physical common sense in video-based VLMs. Moreover, considering that previous evaluation metrics did not specifically

---

[1]The version is gpt4o-0806

target physical correctness, we propose a three-tier hierarchical evaluation strategy for this aspect, transitioning from image-based to comprehensive video analysis: single image, multiple images, and full video stages. Each stage employs distinct VLMs along with custom instructions generated by GPT-4o to form judgments. By combining *PhyGenBench* and *PhyGenEval*, we can efficiently evaluate different T2V models' understanding of physical commonsense at scale, producing results highly consistent with human feedback.

The contributions of our work are three-fold. **i):** We proposed *PhyGenBench*, which compasses a wide range of clear physical phenomenons and explicit physical laws. This benchmark can comprehensively measure whether T2V models understand intuitive physics and indirectly assess their gap from world simulator capabilities **ii):** Along with the benchmark, we propose an automated evaluation framework - *PhyGenEval*, which overcomes the challenges of assessing the correctness of physical commonsense with other metrics and demonstrates high consistency with human feedback on *PhyGenBench*, enabling users to conduct large-scale automated testing of various T2V models. **iii):** We conduct extensive evaluations of popular T2V models, even the best-performing model, Gen-3, scores only $0.51$. This indicates that current models are still far from functioning as world simulators. Based on our evaluation results, we conduct an in-depth analysis and discover that addressing issues such as dynamics is still challenging through prompt engineering or simply scaling up model. We hope this work inspires the community to focus on the learning of physical commonsense in T2V models, rather than merely using them as tools for entertainment.

## 2. Related work

### 2.1. Benchmarks for text-to-video generation

The rapid advancement of text-to-video (T2V) generation models has necessitated various benchmarks for accurate assessment. Traditional works in video generation, such as FVD (Unterthiner et al., 2018), rely on datasets like UCF-101 (Soomro, 2012) and Kinetics-400 (Kay et al., 2017), which are limited in scope. Recent benchmarks, including VBench (Huang et al., 2024) and EvalCrafter (Liu et al., 2024c), aim to comprehensively evaluate general video quality across multiple dimensions. In contrast, some studies focus on fine-grained evaluation of text-to-video (T2V) models from specific aspects. For instance, T2V-CompBench (Sun et al., 2024) assesses compositional generation capabilities, while DEVIL (Liao et al., 2024) evaluates dynamic characteristics of generated videos. Although some research like VideoPhy (Bansal et al., 2024) efforts address the dynamic motions naturalness of video generation, their benchmarks fail to succinctly capture fundamental physical laws. Consequently, most existing works overlook this crucial

aspect, which forms the foundation for realizing a world simulator. To address this gap, we introduce *PhyGenBench*, a benchmark designed to comprehensively measure T2V models' understanding of physical commonsense. We provide more releated works in Appendix A

## 3. PhyGenBench

Inspired by (Swartz, 1985), we first define the following terms: *"Physical Commonsense:"* Basic intuitive understanding of how physical objects and actions behave in everyday life; *"Physical Laws:"* Universal scientific principles that describe consistent behaviors in nature; *"Physical Phenomenon:"* Observable events or processes caused by the interaction of physical laws. The purpose of *PhyGenBench* is to evaluate whether T2V models understand physical commonsense, while each prompt in *PhyGenBench* presents a clear physical phenomenon and an underlying physical law.

**Overview.** As illustrated in Figure 2 (a), *PhyGenBench* encompasses four major categories of physical commonsense: *"Mechanics"*, *"Optics"*, *"Thermal"*, and *"Material Properties"*. It incorporates 27 physical phenomena with intrinsic physical laws reflected by the corresponding designed 160 prompts:

1. *"Mechanics"* covers 7 common mechanical laws: gravity, buoyancy, solid pressure, atmospheric pressure, elasticity, friction, and surface tension, with 40 validated prompts. For example, we use *"A piece of iron is gently placed on the surface of the water in a tank filled with water"* to test T2V model's understanding of Buoyancy, where the iron should sink due to its higher density compared to water.

2. *"Optics"* categorizes 6 aspects based on light phenomena: reflection, refraction, scattering, dispersion, interference & diffraction, and straight-line propagation, yielding 50 prompts. A prompt like *"a kite soaring above a smooth and tranquil pond"* is used to test reflection.

3. *"Thermal"* considers 6 phase transitions: Solidification, Melting, Liquefaction, Boiling, deposition, Sublimation, comprising 30 prompts. Inspired by ChronoMagicBench (Yuan et al., 2024), the vaporization process is evaluated by the prompt *"a timelapse capturing the transformation of water as the temperature rapidly rises above $100°C$"*.

4. *"Material Properties"* includes 5 physical properties (color, hardness, solubility, combustibility, and flame reaction) and 3 chemical properties (acidity, redox potential, and dehydrating properties), resulting in 40 prompts. We reflect material properties, e.g., *"hardness"*, through the prompts with expected phenomena, e.g., *"an egg being hurled with significant force towards a rock"*, where the egg should break while the rock remains intact.

Multiple physical laws could be included in a single prompt, which may bring confusion to the evaluation of physical

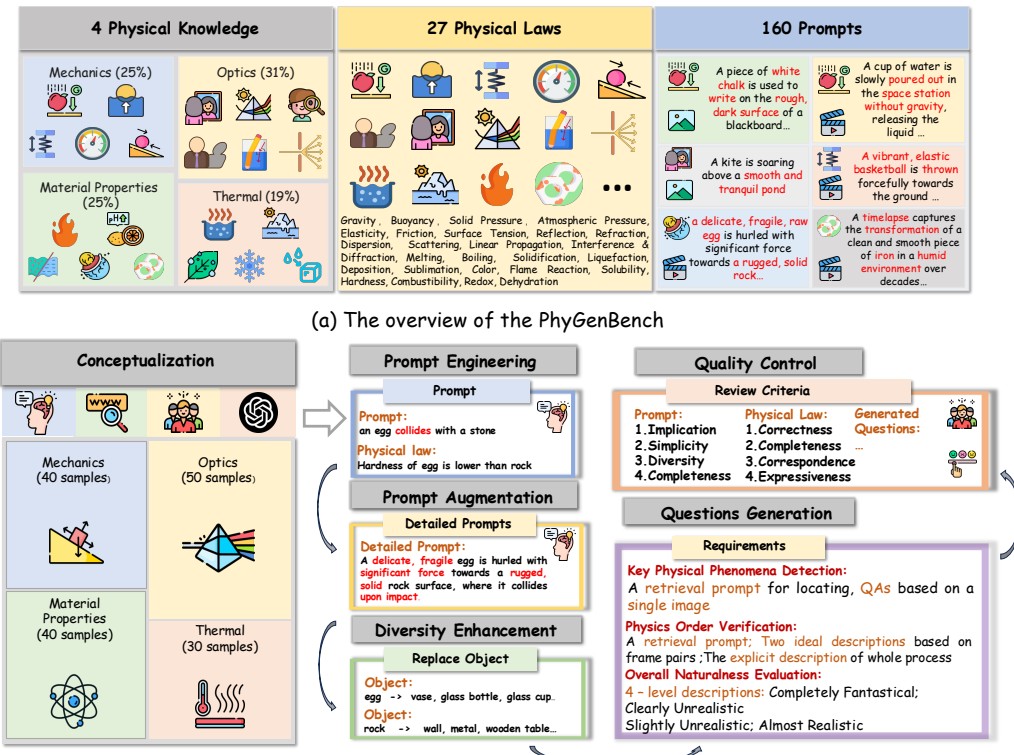

(a) The overview of the PhyGenBench

(b) The data construction pipeline of the PhyGenBench

Figure 2: (a) is the overview of the proposed *PhyGenBench*. (b) is the *PhyGenBench* data pipeline, which covers four physics categories. We select key physical laws and manually craft initial prompts that reflect the corresponding physical phenomena. GPT-4o adds details and enhances diversity by varying objects. Then, we generate different questions for scoring in each evaluation stage. After manual review, we obtain 160 T2V prompts.

common sense in video generation, even for human annotators. To avoid this, we carefully curate prompts to ensure a one-to-one correspondence for each physical phenomenon it reflects, with clear physical law inside. By incorporating physical laws from four distinct physical categories, *PhyGenBench* offers a thorough assessment of current T2V models' understanding of physical commonsense.

**Benchmark Construction.** As shown in Figure 2 (b), we develop a comprehensive approach to create *PhyGenBench*. The methodology encompasses five steps: **1) Conceptualization:** Following (Halliday et al., 2013), We first identify key physical commonsense from four major categories of physics. For each category, we select specific physical laws from textbooks (Harjono et al., 2020), which can be widely recognized and can be easily demonstrated through clear, observable physical phenomenon. **2) Prompt Engineering:** For each physical law, we manually craft the initial T2V prompts to clearly depict the underlying physical phenomenon **3) Prompt Augmentation:** To enhance the model's video generation capabilities, we augment the initial T2V prompts by adding additional details, such as more precise descriptions of objects and actions (Yang et al.,

2024). This augmentation process is carefully designed to avoid revealing the expected physical phenomenon. **4) Diversity Enhancement:** Following T2V-CompBench (Sun et al., 2024), we employ GPT-4o to perform object substitution on the augmented prompts. This step increases the diversity of the benchmark. **5) Question Generation:** Considering that understanding the physical principles implied in prompts increases the evaluation difficulty, following the method of (Singh & Zheng, 2023). For each prompt in *PhyGenBench*, we leverage the world knowledge of LLMs to generate different questions for scoring for different evaluation stage in *PhyGenEval* that reflect physical correctness. These questions are manually reviewed, filtered, and then incorporated into the benchmark. **6) Quality Control:** We conduct a thorough review of the prompts, questions and their associated physical laws to ensure accuracy and relevance. Specifically, we ensure that they are clear and accurate. We then randomly use the current T2V model to check if the prompts are simple enough for the model to generate semantically accurate videos. In this process, we focus on the most fundamental physical laws, the most common scenarios, and rigorous quality standards. We find that excessive prompts are unnecessary: even the most basic

physical laws and scenarios are sufficient to reveal significant issues in current models and effectively distinguish between different models. For more detailed information can be referred to the Appendix B.

# 4. PhyGenEval

*PhyGenEval* aims to assess whether the physical phenomena in the generated videos conform to the corresponding physical laws. To obtain a clear judgment, we decompose the evaluation into semantic alignment (SA) and physical commonsense alignment (PCA). While SA evaluates whether the semantic meaning inferred by the generated video and the input prompt are matched, PCA measures whether the evaluated physical laws are grounded in the videos. For example, for the scene *"an egg collides with a stone"*, SA requires a video containing the egg, the stone, and the collision action. PCA necessitates a video for the whole physical motions in which the egg hits a stone and then breaks, while the stone remains intact. Following (He et al., 2024b), we convert both SA and PCA to a four-point scale, as well as the human ratings.

## 4.1. Semantic Alignment Evaluation

Directly asking the Vision-Language Model(VLM) to align the semantic meaning between videos and input prompts are difficult, as prompts usually are mixed with semantic entities and physical phenomena, and the intermediate outcomes are subtly implied by the videos. For example, in a prompt like *"A timelapse captures the transformation of soup as the temperature rises above 100°C"*, a possible video generation would appear like *"The video shows a soup, but there is no transformation of the soup"*. To address the challenge, we first employ GPT-4o to extract object and action from the original text prompt, we then utilize GPT-4o to sequentially determine the presence of extracted objects in the video and verify the occurrence of specified actions. This decomposition provides more fine-grained captures and prevents the model from confusing semantic and physical correctness during evaluation. Experimental results demonstrate that our automated evaluation method aligns more closely with human judgment and outperforms previous methods (He et al., 2024b; Sun et al., 2024) in *PhyGenBench* (Appendix C.1). In this process, we ensure that GPT-4o's API is configured without randomness, e.g., setting the temperature to 0, to guarantee the reproducibility of the results.

## 4.2. Physical Commonsense Evaluation

To evaluate physical correctness in the video, we evaluated multiple common evaluation metrics comparing human as-

sessments[2]. Experimental results in Table 1 demonstrate that these methods struggle to generalize to the assessment of physical commonsense correctness on *PhyGenBench*, e.g., VideoScore (He et al., 2024b) has only a spearman correlation of 0.19 on *PhyGenBench*, which is most correlated with human assessments except *PhyGenEval*. We attribute it to the main factor: Directly using video-based VLMs fails to comprehend the embedded physical commonsense (Jassim et al., 2023), as current methods are not designed with physical commonsense as a foundation. To fully understand the physical commonsense in the video, there are three key factors need to solve: **i):** Physical processes typically exhibit clear key phenomena depicted by the input prompt (e.g., *"the egg breaks upon hitting the rock."*). It is necessary to identify these key physical phenomena and detect their presence in videos. **ii):** Physical processes are characterized by causality, manifested in the correct sequence of critical events(e.g., *"The egg touchs the rock first, then breaks."*). The correct sequence order validates the correctness of physical processes. **iii):** Physical processes need to possess overall naturalness, which represents the realistic of the overall process. To address these factors, we design a progressive strategy that starts with key physical phenomena, then moves through the sequence of several key phenomena, and finally evaluates the overall naturalness of the entire video. In addition, we use the customized questions designed for each phase in the section 3 to conduct evaluations with different VLMs at each stage. This hierarchical and refined approach reduces the difficulty compared to existing methods that directly uses VLMs to evaluate physical commonsense, enabling *PhyGenEval* to achieve results closely aligned with human judgements. In this process, we ensure that GPT-4o's API and the inference code of VLMs is configured without randomness, to guarantee the reproducibility of the results.

**Key Physical Phenomena Detection.** This stage aims to detect *whether the key physical phenomena occur in the video.* Here we define the key phenomena as an observable and distinctive occurrence (e.g., a specific frame) within a physical process that can directly reveal the corresponding physical law, like deformations or color changes. During the process of constructing the benchmark, for each input prompt in *PhyGenBench*, we craft a retrieval prompt $p_r$ and a set of physics-related questions $Q$, where the retrieval prompt is used to locate the key phenomena frame, and physical-related questions are utilized to check whether the expected physics phenomena are present in the keyframe.

As illustrated in Figure 3 (a), we first obtained both $Q$ and $P_r$ by prompting GPT-4o with the input T2V prompt and corresponding physical law. Following (Hessel et al.,

---

[2]Annotators are asked to score the correctness of physical commonsense in the video. Details refer to Section 5 and Appendix D.1

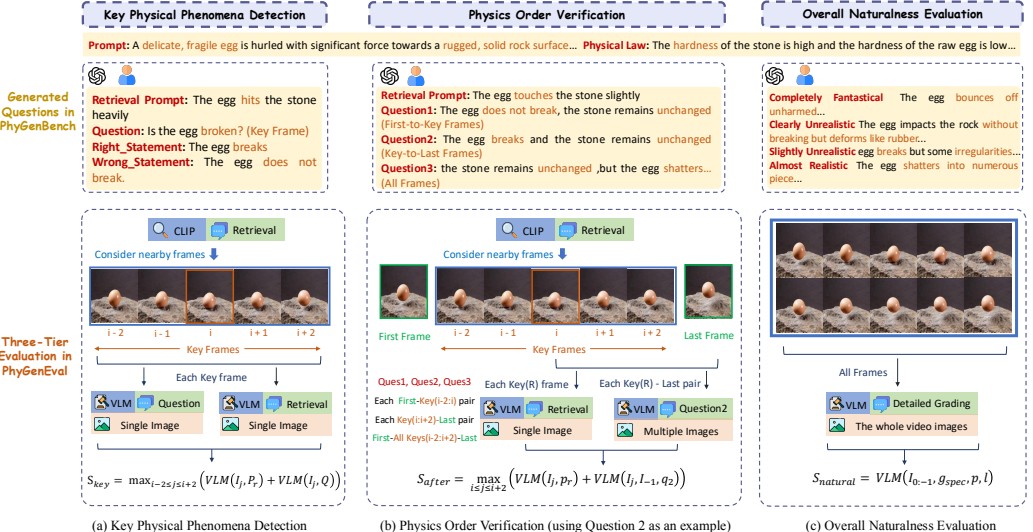

(a) Key Physical Phenomena Detection     (b) Physics Order Verification (using Question 2 as an example)     (c) Overall Naturalness Evaluation

Figure 3: An overview of the proposed *PhyGenEval*. *PhyGenEval* is divided into three parts: Key Physical Phenomena Detection, Physics Order Verification, and Overall Naturalness Evaluation. Each part uses an appropriate VLM in combination with physical-based customized questions generated by GPT-4o. The final score is the combined result of the three parts. For the example in the figure, the three-stage scores are 0, 1 (only $Question_1$ is correct), and 0. The final score is calculated as 0 according to Overall Score calculation in Section 4.2.

2021), a keyframe $I_i$ from the video based on the retrieval prompt, where $I_i$ is the $i$-th frame in the video. By using the keyframe, we define a confidence score of the key phenomena in the video:.

$$S_{\text{key}} = \sum_{q \in Q} \max_{i-2 \leq j \leq i+2} \left( \text{VLM}(I_j, q) + \text{VLM}(I_j, p_r) \right),$$

where $\text{VLM}(I_j, q)$ reflects the presence of physical phenomena in $I_j$ for each related question $q$ from $Q$. $\text{VLM}(I_j, p_r)$ checks whether $I_j$ matches the retrieval prompt, which ensures key phenomena occur at the correct frame. Since videos may contain semantic errors, it's also important for determining if key physical phenomena occur (e.g., an egg shouldn't break in mid-air before hitting a rock). We consider adjacent 5 frames near the keyframe to enhance the robustness. For example, the egg may not be cracked just when it first contacts the stone. We instantiate VLM-based evaluator $\text{VLM}(\cdot)$ with VQAScore (Lin et al., 2024), which has been shown promising evaluation results on visual question-answering.

**Physics Order Verification.** In this stage, we verify *whether key physical phenomena occur in the correct order*. The correct physical sequence is an ordered series of events in a physical process that reflects causality, which represents the necessary prerequisites and temporal order of key physical phenomena. As an example, the egg should first touch the stone and then crack. Considering current models in *PhyGenBench* generally maintain outcome consistency (Huang et al., 2024) (e.g., the egg would not reassemble

itself after it is broken). we approach this direction by investigating the order correctness from the keyframes (Figure 3 (b)), e.g., the keyframe of the egg hits the stone should be ahead of the keyframe of the broken egg.

Similar to the Key Physical Phenomena Detection evaluation, we use a retrieval prompt $p_r$ and three physical-related questions $(q_1, q_2, q_3)$ in *PhyGenBench*. $p_r$ is used to locate the keyframe (e.g., the moment the egg slightly touches the stone.). While $q_1$, $q_2$, and $q_3$ are questions to check the order correctness from the first frame to the keyframe, from the keyframe to the last frame, and from the first frame to the last frame, respectively. Similarly, we first use CLIPScore to locate the key frame $I_i$, then the order correctness scores of $S_{\text{before}}$ and $S_{\text{after}}$ are defined as:

$$S_{\text{before}} = \max_{i-2 \leq j \leq i} \left( \text{VLM}(I_0, I_j, q_1) + \text{VLM}(I_j, p_r) \right)$$

$$S_{\text{after}} = \max_{i \leq j \leq i+2} \left( \text{VLM}(I_j, I_{-1}, q_2) + \text{VLM}(I_j, p_r) \right)$$

$q_3$ assesses the overall physical sequence coherence of the video. The score of answering $q_3$ is defined as by $S_{\text{all}} = \text{VLM}(I_0, I_{i-2:i+2}, I_{-1}, q_3)$, which evaluates the overall sequence (similar to the input video but using manually selected key frames). Here we employ GPT-4o or LLaVA-Interleave (Li et al., 2024) as the VLM-based evaluator $\text{VLM}(\cdot)$, as they demonstrate exceptional multi-image comprehension capabilities. The overall score of whole physical order evaluation can be formulated as $S_{\text{order}} = S_{\text{before}} + S_{\text{after}} + S_{\text{all}}$

**Overall Naturalness Evaluation.** This stage aims to evaluate **the overall naturalness of the video**. we define naturalness as the dynamic progression that aligns with real-world physical phenomenons (Liao et al., 2024). During the construction of *PhyGenBench*, we generate 4 different levels of descriptions $g_{spec}$ for each pair of prompts and their corresponding physical laws. As shown in Figure 3 (c), we require the VLM to score based on $p$, $l$, $g_{spec}$, and the corresponding video denoted by $I_{0:-1}$. Formally, we define the overall naturalness score as:

$$\mathrm{S_{natural}} = \mathrm{VLM}(I_{0:-1}, p, l, g_{spec})$$

We implement the VLM-based evaluator $\mathrm{VLM}(\cdot)$ using InternVideo2 (Wang et al., 2024) and GPT-4o, both of which have promising results in video understanding.

**Overall Score.** We first discretize $\mathrm{S_{key}}$, $\mathrm{S_{order}}$, and $\mathrm{S_{natural}}$ from the three stages into a 4-point scale, then take their average and apply floor rounding as the final score. For robust purposes, we evaluate $\mathrm{S_{order}}$ with both GPT4o and LLaVA-Interleave and $\mathrm{S_{natural}}$ with both GPT4o and InternVideo2. The final score is calculated as the ensemble of two methods. Detailed calculation are provided in Appendix C.

## 5. Experiment

**Experiments Setup.** We evaluate 8 open-source models, as well as 6 proprietary models Kling (kli, 2024), Pika (Pik, 2023), Sora (Sor, 2024), Vidu (Vid, 2024), Hailuo (Hai, 2024), and Gen-3 (gen, 2024). We compare our proposed metric with existing metrics or benchmarks: Videophy (Bansal et al., 2024), VideoScore (He et al., 2024b) and DEVIL (Liao et al., 2024) More Detailed information is provided in Appendix D.

For human evaluation, we compared the results across 8 randomly selected different T2V models. We randomly select 64 prompts from *PhyGenBench* and generate 64 videos for each T2V model. We ask three annotators to provide semantic and physical scores for each video[3]. Each annotator will give an integer score of 0-3 for the semantic and physical scores, and the final score is the average of the three scores and rounded up. Finally, we calculate the correlation between the human scores and automatic evaluation scores using Kendall's $\tau$ and Spearman's $\rho$. we put more detailed information about human evaluation in Appendix D.1.

**Human Evaluation.** As shown in Table 1, current video generation evaluation metrics largely overlook physical correctness. In contrast, *PhyGenEval* implements a detailed design for evaluating physical correctness, demonstrating strong correlations with human judgments across all categories. Its overall correlation coefficient reaches $0.81$,

---

[3]Note that we ask the annotators to focus on the correctness of the physical phenomena for physical scores.

Table 1: **PCA correlation results with proposed *PhyGenEval* in video generation on *PhyGenBench*.** *PhyGenEval* is significantly closer to human feedback on *PhyGenBench* compared to other metrics.

| Metric | Mechanics | | Optics | | Thermal | | Material | | Overall | |
|---|---|---|---|---|---|---|---|---|---|---|
| | $\tau(\uparrow)$ | $\rho(\uparrow)$ | $\tau(\uparrow)$ | $\rho(\uparrow)$ | $\tau(\uparrow)$ | $\rho(\uparrow)$ | $\tau(\uparrow)$ | $\rho(\uparrow)$ | $\tau(\uparrow)$ | $\rho(\uparrow)$ |
| DEVIL (Liao et al., 2024) | 0.16 | 0.16 | 0.03 | 0.03 | 0.10 | 0.11 | 0.27 | 0.29 | 0.17 | 0.18 |
| VideoPhy (Bansal et al., 2024) | 0.00 | −0.03 | −0.15 | −0.14 | 0.08 | 0.08 | 0.13 | 0.14 | 0.03 | 0.04 |
| VideoScore (He et al., 2024b) | 0.18 | 0.20 | 0.07 | 0.08 | 0.14 | 0.15 | 0.14 | 0.15 | 0.17 | 0.19 |
| *PhyGenEval* | **0.72** | **0.75** | **0.76** | **0.77** | **0.73** | **0.75** | **0.81** | **0.84** | **0.78** | **0.81** |

indicating that *PhyGenEval* serves as an effective human-aligned physical commonsense correctness evaluator for *PhyGenBench*. we provide more details in Appendix D.1.

We conduct several case studies to illustrate the differences between various metrics more clearly. As shown in Figure 4, (a) and (f) reveal that VideoScore and DEVIL are prone to misclassifying videos that have smooth and consistent motion but violate fundamental physical laws. Specifically, as for (a), when *"an egg exhibits rubber-like elasticity upon impact with a rock instead of breaking,"* these metrics incorrectly evaluate it as physically correct. VideoPhy exhibits similar limitations. In (c), it incorrectly assesses *"a rock floating on water instead of sinking"* as physically correct. Furthermore, our analysis reveals a major flaw in these three methodologies: they cannot incorporate domain-specific physical commonsense. As illustrated in (e), where *"the flame from burning copper appears red instead of green,"* these metrics fail to identify the mistake. This indicates their inability to incorporate domain-specific physical commonsense. In contrast, *PhyGenEval* demonstrates a robust integration of physical commonsense and comprehensive video content analysis, resulting in more accurate and physically consistent evaluations in *PhyGenBench*.

**Quantitative Evaluation.** We conduct extensive experiments on a wide range of popular video generation models. As illustrated in Table 2, even the best-performing model, Gen-3, only attains a PCA score of $0.51$ on *PhyGenBench*. This indicates that even for prompts containing obvious physical commonsense, current T2V models struggle to generate videos that comply with intuitive physics. It indirectly reflects that these models are still far from achieving the world simulator.

Furthermore, we identify the following key observations: **1):** Across various categories of physical commonsense, all models consistently demonstrate superior performance in the domain of optics compared to other areas. Notably, Vchitect2.0 and CogVideoX-5b achieve a PCA score in the optics domain comparable to that of closed-source models. We posit that this superior performance in the optics domain can be attributed to the abundant and explicit representation of optical knowledge in pre-training datasets, thereby enhancing the model's comprehension in this area. **2):** Sora and

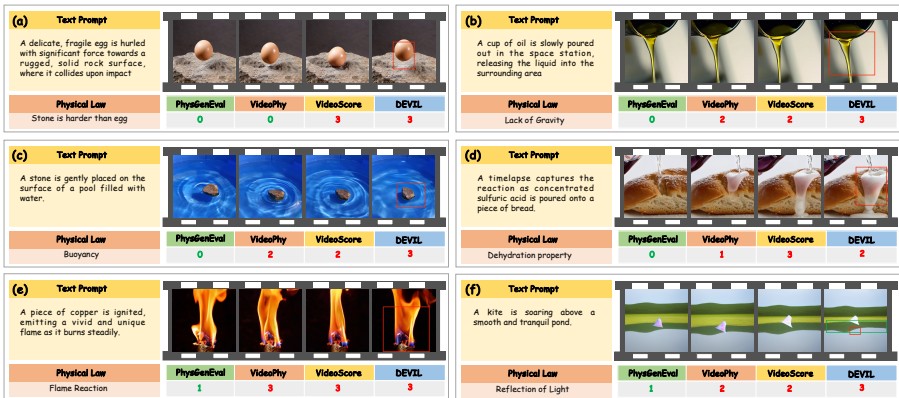

Figure 4: Different video generation evaluation metric in *PhyGenBench*. Except for the proposed *PhyGenEval*, the current methods cannot reasonably assess the correctness of physical commonsense in videos from *PhyGenBench*.

Table 2: **Evaluation results of PCA with the proposed *PhyGenEval* in videos generated by several models** . The results reveal that all models score very low in physical commonsense accuracy. The scores are normalized to a range of 0-1.

| Model | Size | Mechanics(↑) | Optics(↑) | Thermal(↑) | Material(↑) | Average(↑) |
|---|---|---|---|---|---|---|
| CogVideoX (Yang et al., 2024) | 2B | 0.38 | 0.43 | 0.34 | 0.39 | 0.39 |
| CogVideoX (Yang et al., 2024) | 5B | 0.39 | 0.55 | 0.40 | 0.42 | 0.45 |
| Open-Sora V1.2 (Zheng et al., 2024a) | 1.1B | 0.43 | 0.50 | 0.44 | 0.37 | 0.44 |
| Lavie (Wang et al., 2023) | 860M | 0.30 | 0.44 | 0.38 | 0.32 | 0.36 |
| Vchitect 2.0 (Wang et al., 2023) | 2B | 0.41 | 0.56 | 0.44 | 0.37 | 0.45 |
| Hunyuan (Kong et al., 2024) | 13B | 0.44 | 0.53 | 0.38 | 0.39 | 0.47 |
| Pyramid Flow (flux) (Jin et al., 2024) | 2B | 0.37 | 0.50 | 0.47 | 0.37 | 0.43 |
| Pyramid Flow (sd3) (Jin et al., 2024) | 2B | 0.42 | 0.49 | 0.36 | 0.45 | 0.41 |
| Pika1.0 (Pik, 2023) | - | 0.35 | 0.56 | 0.43 | 0.39 | 0.44 |
| Gen-3 (gen, 2024) | - | 0.45 | 0.57 | 0.49 | **0.51** | 0.51 |
| Kling (kli, 2024) | - | 0.45 | 0.58 | 0.50 | 0.40 | 0.49 |
| Sora (Sor, 2024) | - | **0.50** | 0.66 | **0.56** | 0.46 | **0.55** |
| Vidu (Vid, 2024) | - | 0.48 | 0.63 | 0.52 | 0.50 | 0.54 |
| Hailuo (Hai, 2024) | - | 0.49 | **0.67** | 0.50 | 0.50 | **0.55** |

Hailuo exhibit significantly higher performance compared to other models. Specifically, they demonstrate a robust understanding of material properties, achieving a score of 0.55, which substantially surpasses other models. Kling performs particularly well in thermal, attaining the highest score of 0.50 in this domain. **3):** Among open-source models, Hunyuan and CogVideoX 5b perform comparatively well, both exceeding the performance level of Pika. In contrast, Lavie consistently exhibits lower physical correctness. We provide qualitative analysis in Appendix D.3

**Ablation Study.** We conduct a detailed robustness analysis, Experimental results demonstrate that the key designs of *PhyGenEval* are essential, and **even when only using open-source models, *PhyGenEval* maintains high effectiveness**. Detailed results are provided in Appendix D.4.

## 6. Discussion

To explore potential solutions for the challenges posed by *PhyGenBench*, We focus on widely used and proven-effective methods such as scaling laws (Kaplan et al., 2020), prompt engineering (Fu et al., 2024), and some method like

Venhancer (He et al., 2024a) aimed to improve general video quality (Huang et al., 2024). More detailed results are in Appendix E.

**Rewriting prompt.** We aim to explore whether GPT-augmented prompts can address the *PhyGenBench* challenges. Specifically, we rewrite the original prompts using GPT, adding expected physical outcomes and processes. For example, after *"A bottle of juice is slowly poured out in the space station, releasing the liquid into the surrounding area"*, we add *"The liquid forms floating globules, spreading out and moving randomly through the air."* in the end.

As shown in Table 3, we use CogVideoX 5b and Kling as representative models for open-source and closed-source systems, respectively, to conduct tests. The results indicate that prompt rewriting does help the models generate images aligned with physical laws, but it is still far from resolving the issues highlighted by *PhyGenBench*. Both CogVideoX 5b and Kling exhibit some growth, but even for Kling, it only achieves a score of 0.56. This demonstrates that current models still severely lack the ability to

Table 3: **Evaluation results of PCA using the proposed *PhyGenEval* after rewriting prompts** . The results indicate that although using rewritten prompts leads to some improvement, it is still insufficient to address the challenges highlighted by *PhyGenBench*. The scores are normalized to a range of 0-1.

| Model | Size | Mechanics(↑) | Optics(↑) | Thermal(↑) | Material(↑) | Average(↑) |
|---|---|---|---|---|---|---|
| **Before Rewriting Prompt** | | | | | | |
| CogVideoX (Yang et al., 2024) | 5B | 0.39 | 0.55 | 0.40 | 0.42 | 0.45 |
| Kling | - | 0.45 | 0.58 | 0.50 | 0.40 | 0.49 |
| **After Rewriting Prompt** | | | | | | |
| CogVideoX (Yang et al., 2024) | 5B | 0.39 | 0.62 | 0.53 | 0.52 | 0.52 |
| Kling | - | 0.50 | 0.64 | 0.61 | 0.48 | 0.56 |

Table 4: **PCA evaluation results with proposed *PhyGenEval* in videos after VEnhancer**. The results indicate that employing VEnhancer fails to enhance the model's comprehension of physical commonsense. The scores are normalized to a range of 0-1.

| Model | Size | Mechanics(↑) | Optics(↑) | Thermal(↑) | Material(↑) | Average(↑) |
|---|---|---|---|---|---|---|
| Vchitect 2.0 | 2B | 0.41 | 0.56 | 0.44 | 0.37 | 0.45 |
| Vchitect 2.0 (Venhancer) | 2B | 0.41 | 0.56 | 0.42 | 0.38 | 0.45 |

accurately render physical scenes, and this deficiency cannot be easily resolved through simple prompt rewriting. To illustrate this issue more clearly, as shown in Figure 10, our qualitative analysis shows that rewriting prompts can only address simple issues (e.g., flame color reactions), but remains ineffective for more complex physical processes (e.g., egg breaking, stone sinking).

**The robustness of *PhyGenBench* and *PhyGenEval*.** VEnhancer (He et al., 2024a) is a generative space-time enhancement framework that improves existing videos by adding spatial details and synthetic motion in the temporal domain. After enhancement by VEnhancer, Vchitect2.0 shows significant improvement on VBench, even surpassing Kling. However, VEnhancer only enhances the visual quality of videos (e.g., making them more coherent and clear) without addressing the model's poor understanding of physical commonsense.

As shown in Table 4, Vchitect enhanced by VEnhancer still scores similarly to the original version on *PhyGenBench*. We calculate a high Spearman coefficient of 0.86 between model scores on *PhyGenBench* before and after VEnhancer enhancement. This indicates that *PhyGenEval* primarily focuses on physical correctness and is robust to other factors affecting visual quality. Furthermore, it demonstrates that even if a model can generate videos with better general quality (e.g., ranking higher on VBench), it doesn't necessarily imply a better understanding of physical common sense. This highlights the distinction between *PhyGenBench* and

benchmarks like VBench that evaluate video quality.

## 7. Conclusion

In this paper, we explore the gap between current T2V models' understanding of physical commonsense and their role as world simulators. To achieve this, we introduce *PhyGenBench* and *PhyGenEval*. *PhyGenBench* is a benchmark specifically designed to assess models' understanding of physical commonsense, featuring various physical laws and simple, clear physical phenomenons. Alongside *PhyGenBench*, we propose a novel evaluation framework called *PhyGenEval* to automate the evaluation process. Experimental and analytical results show that current T2V models struggle to generate videos that align with physical commonsense, highlighting a significant gap from world simulation.

## Impact Statement

This work contributes to the long-term vision of advancing video generation models toward more physics-consistent world simulators. From an ethical perspective, while strict physical consistency evaluation may reveal current model limitations, it also opens new directions for improving models' physical understanding, data synthesis, and evaluation frameworks. The future societal impact includes enhancing the reliability of video generation technology in scientific simulation, education, and filmmaking.

## Acknowledgements

This paper is partially supported by the National Key R&D Program of China No.2022ZD0161000.

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

# A. Related work

## A.1. Benchmarks For Physical Understanding

Recent benchmarks such as SuperCLEVR-Physics (Li et al., 2023), ContPhy (Zheng et al., 2024b), and Physion (Bear et al., 2021) have significantly advanced the evaluation of AI's physical reasoning, focusing on understanding and predictive tasks. SuperCLEVR-Physics emphasizes reasoning about dynamic properties like velocity and collisions in 4D scenes, ContPhy expands the evaluation scope to include diverse physical properties, such as mass and density, within continuum settings. It underscores the limitations of existing AI models in handling soft-body dynamics. And Physion evaluates models' ability to predict physical phenomena like collisions and motion while benchmarking against human behavior. However, these works primarily target understanding or prediction rather than generative capabilities. In contrast, our work introduces PhyGenBench, a comprehensive benchmark designed to evaluate whether Text-to-Video (T2V) models can generate videos that adhere to physical commonsense. Unlike existing benchmarks, our work highlights the generative challenges in intuitive physics, revealing critical gaps in current T2V models and underscoring the need to advance physical commonsense for applications beyond entertainment.

## A.2. Evaluation metrics for text-to-video generation

Conventional approaches to video quality assessment often employ metrics such as FVD (Unterthiner et al., 2018) and IS (Salimans et al., 2016). However, the detection of unrealistic motions is difficult for them (Brooks et al., 2022), and FVD requires a reference video that is hard to obtain for novel scenes, making it challenging to evaluate the correctness of physical commonsense. Recent studies have explored the use of advanced vision-language models (VLMs) as evaluators. For instance, VideoScore (He et al., 2024b) leverages human feedback to train models for video quality assessment, while T2V-CompBench (Sun et al., 2024) utilizes powerful models like LLaVA (Liu et al., 2024a) to evaluate the correctness of spatial relationships. Although a few works demonstrate improved alignment with human judgments, they fall short in generalizing to assessments of physical commonsense correctness. To address this limitation, we introduce *PhyGenEval*, a novel method designed to evaluate physical commonsense correctness on *PhyGenBench*. We validate the efficacy of our approach through comprehensive human correlation studies.

# B. PhyGenBench Details

## B.1. Detailed Overview

A fine-grained analysis of the dataset is essential for a comprehensive understanding of the benchmark. As shown in Table 5, *PhyGenBench* covers 4 major domains in physics, encompassing 27 representative physical laws, which enables it to provide a more comprehensive and fine-grained evaluation of models' physical capabilities. We evaluate 14 advanced models including Sora, Vidu, and etc. Additionally, our captions encompass totally 165 unique objects and 42 unique actions with an average length of 18.75 words.

Table 5: Details of *PhyGenBench*

| Statistic | Number |
|---|---|
| Physical Laws | 27 |
| Domains | 4 |
|    Optics | 50 |
|    Mechanics | 40 |
|    Thermal | 30 |
|    Material Properties | 40 |
| Total Captions | 160 |
| Total T2V Models | 14 |
| Total Generated Videos | 2240 |
| Unique Objects | 165 |
| Unique Actions | 42 |
| Average Length of Caption | 18.75 |

## B.2. Construction

we provide additional details about the Questions Generation and Quality Control processes described in Section 3.

**Questions Generation.** Considering that a single prompt cannot fully reflect expected physical laws—such as placing a wooden block on water, where buoyancy should cause it to float rather than sink—direct evaluation of videos by VLMs (Vision-Language Models) using prompts alone is challenging. Research indicates that VLMs struggle to directly comprehend physical laws inherent in videos (Chow* et al., 2025). Therefore, we leverage the extensive world knowledge of LLMs (Large Language Models) to parse prompts and generate physics-informed questions for different stages of evaluation (as detailed in *PhyGenEval*). This approach enables

VLMs to accurately determine the physical correctness of videos. We have provided the complete prompt and code in our anonymous link. Specifically, for different stage in *PhyGenEval*, we generate different questions for scoring. **In Key Physical Phenomena Detection:** At this stage, we detect key frames that contain the expected physical phenomena. For each input prompt in *PhyGenBench*, we design a retrieval prompt $p_r$ and a set of physics-related questions $Q$. The retrieval prompt is used to locate the key frame, while the questions are used to verify whether the expected physical phenomena are present in the identified frame. For example, when parsing the prompt "an egg colliding with a stone," we use GPT-4o with the prompt: "Propose 1-2 questions with answers based on the given prompt and physical laws. Include a retrieval prompt for identifying the key frame, and ensure the question focuses on a single image.". **In Physics Order Verification:** This stage examines whether the sequence of physical events is correct, e.g., whether the egg hits the stone before breaking. We evaluate this by checking the order correctness between key frames (Figure 3 (b)). For example, the frame showing the egg hitting the stone should precede the frame showing the broken egg. Similarly, we prompt GPT-4o to generate a retrieval prompt $p_r$ and three physics-related questions $(q_1, q_2, q_3)$. The retrieval prompt identifies the key frame, while $q_1$, $q_2$, and $q_3$ verify the order correctness from the first frame to the key frame, from the key frame to the last frame, and from the first frame to the last frame, respectively. The GPT-4o prompt we use is: "Provide the retrieval prompt and two ideal descriptions based on the selected frame pairs (first-retrieval and last-retrieval) as well as a complete description of the process." **In Overall Naturalness Evaluation:** This stage evaluates the overall physical correctness of the dynamic process. For each prompt-physical law pair, we require GPT-4o to generate four levels of descriptions reflecting increasing correctness: Completely Fantastical, Clearly Unrealistic, Slightly Unrealistic, and Almost Realistic. The GPT-4o prompt we use is: "Evaluation Standards: Completely Fantastical: Entirely detached from reality, featuring elements of fantasy or surrealism. Clearly Unrealistic: Contains significant and sustained violations of physical laws, such as implausibly large objects or scenes. Slightly Unrealistic: Features minor or brief distortions, such as unnatural facial expressions or textures that are difficult to notice. Almost Realistic: No noticeable distortions; fully consistent with reality."

**Quality Control.**    This process is primarily conducted through manual checks. Specifically, we perform detailed evaluations during the benchmark construction process, including verifying the correctness of prompts and their associated physical laws, assessing the semantic simplicity of prompts, and ensuring the accuracy of generated questions. Five senior undergraduate students are recruited, with each question assigned to all five annotators for evaluation.

For each prompt, annotators first verify the correctness of the prompt and its corresponding physical law, cross-checking the information using online resources and tools. Next, for semantic simplicity, we refine the benchmark by removing overly complex prompts that current models cannot reasonably handle. Annotators also assess whether the T2V-generated videos are semantically reasonable to ensure they support effective evaluation. Finally, for the questions generated in Section 3 used for scoring, annotators assess the physical correctness of each GPT-generated question. For the Overall Naturalness Evaluation stage, the criteria require that each level of description shows progressive improvements in correctness and distinguishability.

Through this process, we optimize the dataset quality based on annotator feedback, producing a high-quality dataset that includes prompts, their corresponding physical laws, and the questions used for scoring during evaluation.

### B.3. Difference between Videophy and Ours

VIDEOPHY (Bansal et al., 2024) comprises 688 curated simple prompts that focus on interactions between three types of physical materials: solid-solid, solid-fluid, and fluid-fluid, but lack annotations of physical laws. The dataset is designed to evaluate a model's understanding of physical commonsense, featuring a limited range of physical phenomenons such as rigid body interactions, fluid dynamics, and contact forces. We are better suited than Videophy for evaluating physical commonsense due to two significant differences.

First As shown in Figure 2, *PhyGenBench* includes 160 carefully crafted prompts across 27 distinct physical laws, spanning four fundamental domains, which comprehensively assess a model's understanding of physical commonsense. While Videophy primarily focuses on interactions between solid-fluid, solid-solid, and fluid-fluid, limiting its coverage and overlooking common physical laws such as phase transitions and basic material properties. What' more, Videophy lacks annotations of physical laws making it hard for VLM model to evaluate. Second, as shown in Table 6, the average SA score of *PhyGenBench* (0.80) significantly outperforms that of Videophy (0.63). This indicates that *PhyGenBench* prompts are well-suited and easy for T2V models to generate high-quality, well-aligned videos, which benefits evaluation of physical correctness. In contrast, as shown in Figure 5, We find that prompts from Videophy pose challenges for T2V models in generating text-aligned and high-quality videos for two main reasons: 1. The prompts lack detail and specificity. For

instance,*"A tissue blots a tear from an eye"* is overly simplistic (without augmentation). Modern T2V models, such as CogVideo5B ([Yang et al.](), [2024]()), are typically trained with longer and more descriptive captions, which enhance their ability to comprehend and generate content based on prompts. 2. The scenes are often complex and unrealistic. For example, "The wristwatch knob winds the inner spring tightly" describes a process involving intricate internal mechanisms that are not visible externally. As a result, it is exceedingly difficult for T2V models to generate such scenes accurately.

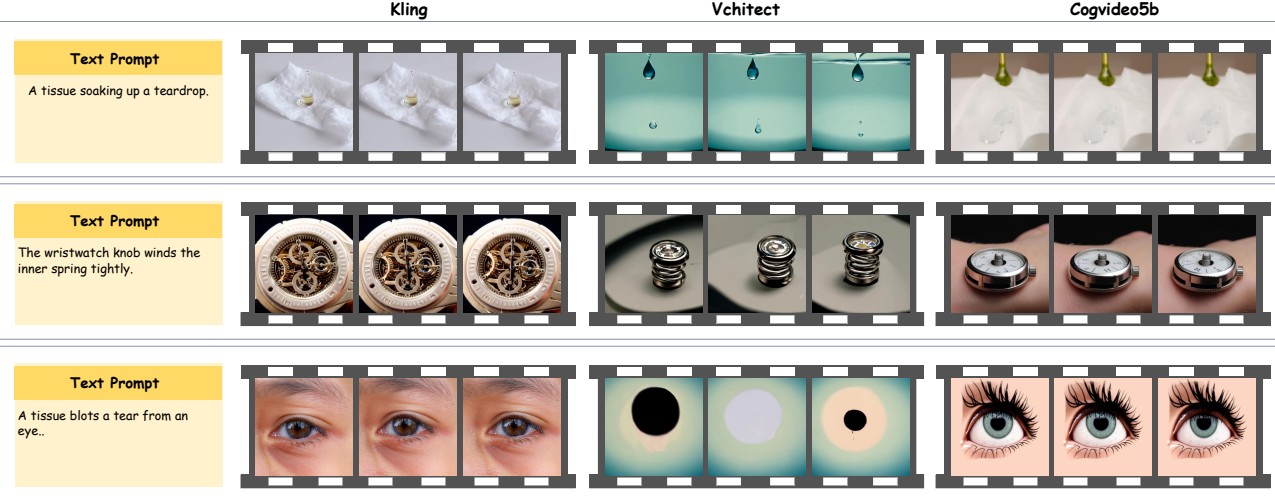

Figure 5: **Samples of videos generated by Kling, Vchitect, and Cogvideo5b in Videophy.** All T2V models struggle to achieve proper text alignment and produce high-quality videos, making it meaningless to evaluate physical correctness in Videophy.

Table 6: **Comparison of SA results for video generation between Videophy and *PhyGenBench*.** We randomly select 64 prompts from both Videophy and *PhyGenBench*, use different T2V models to generate videos, and then ask annotators to score based on our cretiera in Figure [11](). The results show that *PhyGenBench* 's SA scores significantly outperform Videophy.

| Model | Size | Videophy(↑) | *PhyGenBench* (↑) |
|---|---|---|---|
| CogVideoX ([Yang et al.](), [2024]()) | 5B | 0.48 | 0.78 |
| Vchitect 2.0 | 2B | 0.63 | 0.84 |
| Kling | - | **0.77** | **0.89** |
| Average | - | 0.63 | 0.80 |

## C. PhyGenEval Details

### C.1. Semantic alignment details

To reduce the complexity for VLM models to evaluate semantic correctness of generated videos between prompts, we adopt a two-stage strategy. Initially, we employ GPT-4o to extract objects and actions from the original text prompt. Subsequently, we employ GPT-4o to determine whether the extracted objects are present in the video and to verify the occurrence of specified actions. For each video, GPT-4o first assesses the presence of the objects mentioned in the prompt (e.g., *"egg"*) within the video frames. This evaluation is performed according to Question 1 (Q1), where GPT-4o assigns a score from 0 to 2 based on the completeness of object presence: a score of 2 is given if all the objects are present, 1 if some of the objects are missing, and 0 if none of the objects appear in the video. After determining object presence, GPT-4o moves on to Question 2 (Q2) to check if the specified action (e.g., *"pour out"*) is performed in the video. It assigns a binary score (0 or 1) depending on whether the action is present (1) or absent (0). Finally, these scores are combined to form the overall semantic alignment score. we put more details about other metric baselines in Appendix [D.1](). **In this process, we ensure that GPT-4o's API is configured without randomness, e.g., setting the temperature to 0, to guarantee the reproducibility of the results.**

## C.2. Physical Commonsense alignment details

In this section, we use the same notation as in Section 4.2 and provide a more detailed description of the calculation and design of the method. **In this process, we ensure that GPT-4o's API and the inference code of VLMs is configured without randomness, e.g., setting the temperature to 0, to guarantee the reproducibility of the results.**

**Key Physical Phenomena Detection.** We categorize the T2V prompts into monotonic processes (eg. *"melting with increasing temperature"*) and non-monotonic processes (eg. *"an egg hitting a rock"*) based on the physical phenomena they represent. For prompt with monotonic processes, we only consider using the *"Last Frame"* as the retrieval prompt, resulting in a single question. We can directly calculate $\text{VLM}(\text{Img}_j, Q)$, where the score for the corresponding video of this prompt ranges from 0 to 1. For prompt with non-monotonic processes, we consider both the intermediate key frames and the Last Frame, resulting in two questions. For the intermediate key frames, we calculate $\text{VLM}(\text{Img}_j, Q) + \text{VLM}(\text{Img}_j, P_r)$, which ranges from 0-2. Consequently, the score range for videos corresponding to this prompt is 0 to 3.

For specific calculatation, we need to calculate $\text{VLM}(\text{I}_j, p_r)$ and $\text{VLM}(\text{I}_j, q)$, where $\text{Img}_j$ is the $j$-th frame in the video. For $\text{VLM}(\text{I}_j, p_r)$, the calculation involves assessing the matching degree between the key frame and the retrieval prompt, which can be directly obtained using the original calculation method in (Lin et al., 2024). For $\text{VLM}(\text{I}_j, q)$, we follow the computation approach from ChronoMagicBench (Yuan et al., 2024), we derive $\text{VLM}(\text{I}_j, q)$ by determining the ratio of the VQAScore for the affirmative statement to the combined VQAScores for both the affirmative and negative statements. We perform the calculations of $\text{VLM}(\text{I}_j, p_r)$ and $\text{VLM}(\text{I}_j, q)$ for each key frame within the specified range to obtain the physical correctness score for the problem.

**Physics Order Verification.** For this stage, which we've primarily introduced in Section 4, we focus on key calculation points. The score calculation formula for $q_1$ is $\text{S}_{\text{before}} = \max_{i-2 \leq j \leq i} (\text{VLM}(\text{I}_0, \text{I}_j, q_1) + \text{VLM}(\text{I}_j, p_r))$. Here, $\text{VLM}(\text{I}_j, p_r)$ determines if the retrieved key frame satisfies the retrieval prompt,as the physical phenomenon should occur in the keyframe primarily located in Key Phenomena Detection, which is crucial for Key Sequence Verification (e.g.the expected physical phenomenon of egg cracking should occur in the keyframe when the egg hits the stone, rather than other frames when the egg is in the air or else). $\text{VLM}(\text{I}_0, \text{I}_j, q_1)$ assesses the correctness of the Key Sequence order in the video. Notably, we calculate $\text{VLM}(\text{I}_j, p_r)$ using VQAScore, yielding a decimal between 0 and 1, while $\text{VLM}(\text{I}_0, \text{I}_j, q_1)$ employs VLM (GPT-4V or LLaVA-Interleave) for question-answering, scoring 1 or 0 based on the model's Yes or No response.

**Overall Naturalness Evaluation.** Here we mainly explain how to get the score of this part based on the evaluation results under the two-stage strategy described in Section 4. Specifically, we ask the video-based VLM to select the most appropriate option for the video according to the detailed scoring criteria generated by the LLM, and then we map the options to scores (Completely Fantastical to Almost Realistic corresponds to 0-3 points)

**Overall Score.** We detail the discretization and calculation process of the scores here. In the stage of key phenomena detection, we categorize the prompts into monotonic and non-monotonic processes based on the physical phenomena they represent. For monotonic processes, the score range is 0-1, which we directly discretize by averaging into integer values from 0-3. Specifically, for non-monotonic processes with a score range of 0-3, we discretize the scores to $[1, 1.5, 2.25]$. This is because no points should be awarded if the physical phenomena are incorrect ($\text{VLM}(\text{I}_j, p_r) = 1$ and $\text{VLM}(\text{I}_j, q) = 0$), even with accurate retrieval. (e.g., The egg hits the stone and does not break)

In the stage of key sequence verification, we have three multi-image problems. One point is awarded for each correct answer, resulting in a final integer score from 0-3. Similar to the stage, of key phenomena detection we need to consider both the accuracy of key frame retrieval and the physical question answering. Therefore, we design the following: for $Q_1$, when $\max_{i-2 \leq j \leq i} (\text{VLM}(\text{I}_0, \text{I}_j, q_1) + \text{VLM}(\text{I}_j, p_r))$ and $\text{VLM}(\text{I}_j, p_r) > 0.5$, the question is considered correct. The process for $q_2$ is similar. For $q_3$, it is marked correct when $\text{VLM}(\text{I}_0, \text{I}_{i-2:i+2}, \text{I}_{-1}, q_3)$.

In the stage of overall naturalness evaluation, as we require video-based direct option selection, choosing Completely Fantastical, Clearly Unrealistic, Slightly Unrealistic, and Almost Realistic is scored as 0, 1, 2, and 3 points respectively. Finally, we average all scores and round down to obtain the final score.

For the ensamble operation, in order to reduce the bias caused by using a single VLM at a certain stage, we ensemble the results of PhyGenEval using open source models or closed source models. Specifically, we average the two results and round them down.

# D. Experiment

## D.1. Experiments Setup

**T2V model Implementation details.** Open-Sora 1.2 (Zheng et al., 2024a) is an open-source project with the goal of reproducing Sora. CogVideoX 2b (Yang et al., 2024) and CogVideoX 5b are large-scale diffusion transformer models for text-to-video generation, incorporating a 3D Variational Autoencoder (VAE) for efficient video compression and an expert transformer with Expert Adaptive LayerNorm to improve text-video alignment. LaVie (Wang et al., 2023) is a cascaded video latent diffusion model. Vchitect2.0 (Wang et al., 2023), developed by the Shanghai AI Lab, is an advanced video generation model featuring a Parallel Transformer architecture to scale up video diffusion models and empower video creation. Hunyuan (Kong et al., 2024) is the largest open-source T2V model. Pyramid Flow is a T2V model with autoregressive architecture. Sora (Sor, 2024), Pika1.0 (Pik, 2023), Gen-3 (gen, 2024), Kling (kli, 2024), Vidu (Vid, 2024), and Hailuo (Hai, 2024) are the most popular closed-source models.

**Evaluation Metrics details.** We compare our proposed *PhyGenEval* with some evaluation metrics from previous methods like VideoPhy (Bansal et al., 2024) and VideoScore (He et al., 2024b). VideoPhy fine-tunes a VLM with the VIDEOPHY dataset proposed by themselves, which includes human feed back about the semantic alignment and dynamic motion correctness about videos. VideoScore is trained on the VIDEOFEEDBACK dataset proposed by themselves, Initialized from the Mantis model. VideoScore provides automatic assessments of video quality based on human scoring criteria. To compare with *PhyGenEval* on SA and PCA, We only choose the text alignment and fact consistency criteria. Specifically, for the semantic alignment evaluation, we compare the Grid-LLaVA method proposed by T2V-CompBench, which extends the LLaVA (Liu et al., 2024a) model to handle multi-frame inputs by sampling 6 frames uniformly from a video to create an image grid. For the physical commonsense alignment evaluation, we also compare with DEVIL (Liao et al., 2024), which uses Gemini 1.5 Pro (Reid et al., 2024) to assess the overall naturalness of videos and applies the same scoring standard prompt to all videos.

Furthermore, to evaluate the effectiveness of our *PhyGenEval* designs, we conduct a large amount of ablation studies and pue more details in Appendix D.4.

**Human evaluation details.** Here, we provide a detailed explanation of the human evaluation described in Section 5. Specifically, we require annotators to score based on the standards outlined in Figure 11, covering both semantic alignment and physical commonsense alignment. For example, as for the video shown in Figure 11, The egg bounces off the rock like a rubber ball, completely violating physical laws like dynamics, the annotator gives a score of 0 for physical commonsense alignment. However, since the video fully includes the egg, the rock, and the collision action, the annotator gives a score of 3 for semantic alignment.

## D.2. Quantitative Evaluation

**Comparison result about semantic alignment.** Here we design a new baseline *PhyGenEval* (Grid-LLaVA) to illustrate the superiority of the method, which uses the two-stage strategy proposed in *PhyGenEval* from Appendix C.1, but replaces the VLM with Grid-LLaVA proposed in T2V-CompBench (Sun et al., 2024). As shown in Table 8, *PhyGenEval* achieves the highest correlation scores across all categories, demonstrating its effectiveness as a human-aligned semantic commonsense correctness evaluator for *PhyGenBench*. Compared to other methods, *PhyGenEval* consistently outperforms previous baselines like VideoPhy, VideoScore, and Grid-LLaVA. Specifically, *PhyGenEval* obtains an overall Kendall's $\tau$ of $0.53$ and a Spearman's $\rho$ of $0.56$, surpassing the Grid-LLaVA ($\tau$: $0.35$, $\rho$: $0.39$). The results clearly show that our *PhyGenEval* design provides a more accurate and reliable semantic commonsense evaluation in *PhyGenBench*.

**Quantitative result about semantic alignment.** As shown in Table 9 , nearly all models achieve relatively high SA scores, whether evaluated by machines or humans. This suggests that the scenarios in *PhyGenBench* are relatively straightforward, making it easier to assess physical commonsense. Among all the models, Kling achieved the highest SA score, with a human evaluation score of $0.89$, reflecting its strong instruction understanding and video generation capabilities.

## D.3. Qualitative Evaluation.

The different video cases for $4$ physical commonsense categories are illustrated in Figure 6. Our main observations are as follows: In mechanics, the models struggle to generate simple physically accurate phenomenons. As shown in Figure 6,

Table 7: Details about evaluation models. The table shows duration, FPS, and resolution for each model.

| Model | Duration (s) | FPS | Resolution |
|---|---|---|---|
| Open-Sora 1.2 (Zheng et al., 2024a) | 4 | 24 | 1280 × 720 |
| CogVideoX 2b | 6 | 8 | 720 × 480 |
| CogVideoX 5b | 6 | 8 | 640 × 360 |
| Lavie | 4 | 8 | 512 × 320 |
| Vchitect2.0 | 5 | 8 | 768 × 432 |
| Hunyuan | 5 | 24 | 1280 × 720 |
| Pyramid Flow(flux) | 24 | 8 | 640 × 384 |
| Pyramid Flow(sd3) | 24 | 8 | 640 × 384 |
| Pika (Pik, 2023) | 3 | 24 | 1280 × 720 |
| Gen-3 (gen, 2024) | 11 | 24 | 1280 × 768 |
| Kling (kli, 2024) | 5 | 30 | 1280 × 720 |
| Vidu (Vid, 2024) | 5 | 30 | 1280 × 720 |
| Hailuo (Hai, 2024) | 5 | 30 | 1280 × 720 |
| Sora (Sor, 2024) | 5 | 30 | 1280 × 720 |

Table 8: **SA correlation results with proposed *PhyGenEval* in video generation**. A higher score indicates better performance for a category. **Bold** stands for the best score,

| Metric | Mechanics | | Optics | | Thermal | | Material | | Overall | |
|---|---|---|---|---|---|---|---|---|---|---|
| | $\tau(\uparrow)$ | $\rho(\uparrow)$ | $\tau(\uparrow)$ | $\rho(\uparrow)$ | $\tau(\uparrow)$ | $\rho(\uparrow)$ | $\tau(\uparrow)$ | $\rho(\uparrow)$ | $\tau(\uparrow)$ | $\rho(\uparrow)$ |
| VideoPhy (Bansal et al., 2024) | 0.20 | 0.25 | 0.03 | 0.03 | 0.20 | 0.24 | 0.18 | 0.22 | 0.13 | 0.17 |
| VideoScore (He et al., 2024b) | 0.14 | 0.16 | −0.13 | −0.14 | 0.23 | 0.02 | 0.02 | 0.02 | 0.05 | 0.05 |
| Grid-LLaVA (Sun et al., 2024) | 0.39 | 0.43 | 0.45 | 0.49 | 0.30 | 0.33 | 0.22 | 0.26 | 0.35 | 0.39 |
| *PhyGenEval* (Grid-LLaVA) | 0.35 | 0.38 | 0.46 | 0.48 | 0.41 | 0.44 | 0.42 | 0.45 | 0.42 | 0.44 |
| *PhyGenEval* | 0.48 | 0.52 | 0.64 | 0.67 | 0.46 | 0.49 | 0.47 | 0.50 | 0.53 | 0.56 |

all models fail to depict the glass ball sinking in water. As for (b), instead showing it floating on the surface, OpenSora and Gen-3 even produce videos where the ball is suspended. Additionally, the models do not capture special physical phenomenonss, such as the state of water in zero gravity, as seen in (a). In optics, the models perform relatively better. (c) and (d) show the models handling reflections of balloons in water and colorful bubbles, though OpenSora and CogVideoX still produce reflections with noticeable distortions in (d). In thermal, the models fail to generate accurate videos of phase transitions. For the melting phenomenon in (e), most models show incorrect results, with CogVideoX even producing a video where the ice cream increases in size. Similar errors appear in the sublimation process in (f), with only Gen-3 showing partial understanding. Regarding material properties, (g) shows all models failing to recognize that an egg should break when hitting a rock, with Kling displaying the egg bouncing like a rubber ball. For simple chemical reactions, such as the black bread experiment in (h), none of the models demonstrate an accurate understanding of the expected reaction.

### D.4. Ablation study

**The Component in *PhyGenEval* on physical commonsense alignment evaluation.** We conduct a series of ablation studies to demonstrate the necessity of our method design by examining its correlation with human evaluation results, similar to those described in Section 5. Specifically, we compare: 1) ) The effect of the various stages of *PhyGenEval*, as proposed in Section 4.2; 2) Performance differences when using various VLMs and their ensembles in *PhyGenEval*, as outlined in Section 4.2. 3) The larger open models in PhyGenEval. Notice that *PhyGenEval* for physical commnonsense alignment evaluation consists of three stages: key phenomena Detection, key sequence verification, and overall naturalness evaluation.

Table 9: **SA evaluation results with proposed *PhyGenEval* in video generation**. Both machine and human evaluations indicate that most models achieve good semantic scores on *PhyGenBench*. This suggests that the scenarios in *PhyGenBench* are simple enough to clearly reflect physical phenomena. The scores are normalized to a range of 0-1.

| Model | Size | Mechanics(↑) | Optics(↑) | Thermal(↑) | Material(↑) | Average(↑) |
|---|---|---|---|---|---|---|
| CogVideoX (Yang et al., 2024) | 2B | 0.63 | 0.67 | 0.61 | 0.63 | 0.64 |
| CogVideoX (Yang et al., 2024) | 5B | 0.78 | 0.88 | 0.78 | 0.64 | 0.78 |
| Open-Sora V1.2 (Zheng et al., 2024a) | 1.1B | 0.73 | 0.85 | 0.82 | 0.73 | 0.79 |
| Lavie (Wang et al., 2023) | 860M | 0.47 | 0.63 | 0.73 | 0.53 | 0.58 |
| Vchitect 2.0 (Wang et al., 2023) | 2B | **0.92** | **0.89** | 0.77 | 0.74 | **0.84** |
| Hunyuan (Kong et al., 2024) | 13B | 0.75 | 0.83 | 0.81 | **0.75** | 0.79 |
| Pyramid Flow (flux) (Jin et al., 2024) | 2B | 0.71 | 0.83 | 0.75 | 0.63 | 0.68 |
| Pyramid Flow (sd3) (Jin et al., 2024) | 2B | 0.71 | 0.82 | **0.83** | 0.56 | 0.75 |
| Pika (Pik, 2023) | - | 0.63 | 0.81 | 0.73 | 0.69 | 0.72 |
| Gen-3 (gen, 2024) | - | 0.84 | 0.93 | 0.82 | 0.78 | 0.85 |
| Kling (kli, 2024) | - | 0.88 | 0.91 | 0.87 | 0.74 | 0.85 |
| Sora (Sor, 2024) | - | 0.85 | 0.83 | 0.86 | 0.77 | 0.84 |
| Vidu (Vid, 2024) | - | 0.88 | **1.00** | **0.92** | **0.88** | **0.92** |
| Hailuo (Hai, 2024) | - | **0.94** | 0.98 | 0.86 | 0.79 | 0.87 |

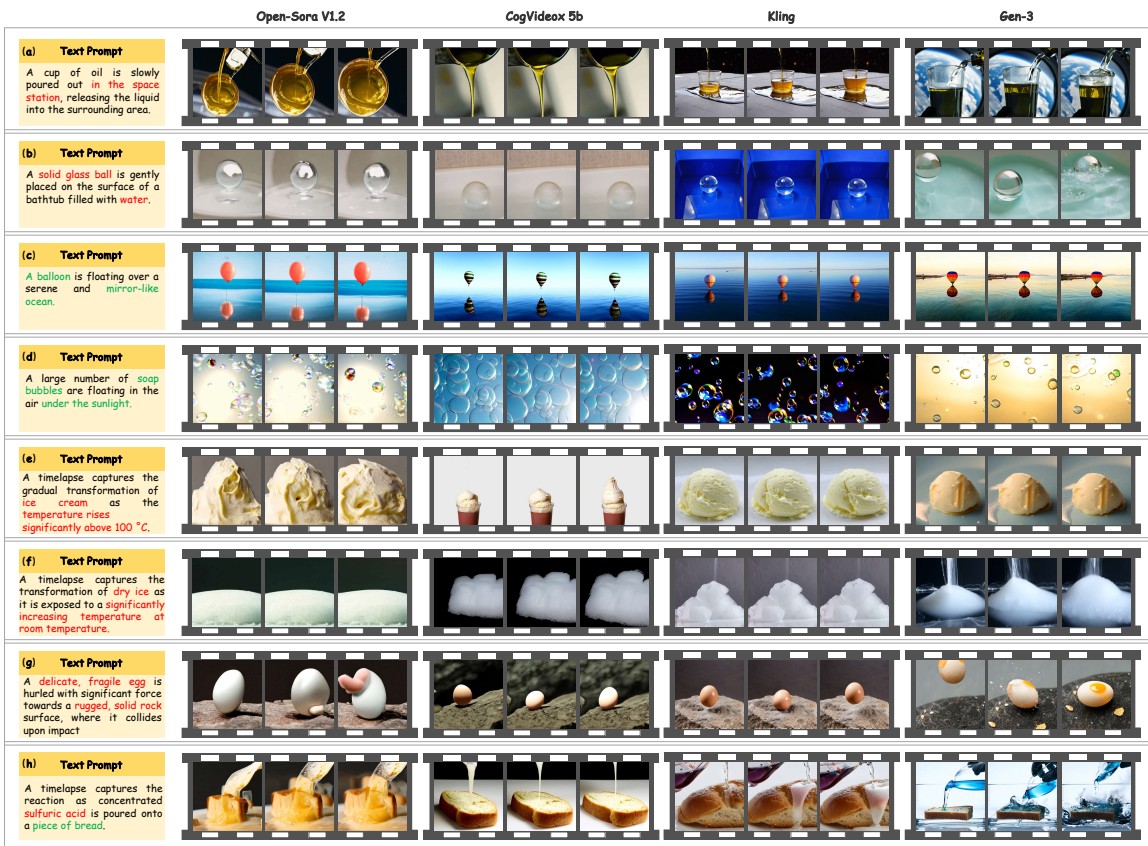

Figure 6: Qualitative comparisons of four categories. Current models perform relatively well in generating optical phenomenons but are weaker in mechanics, thermal, and material properties.

And We denote them as *PhyGenEval*-S, *PhyGenEval*-M, and *PhyGenEval*-V based on the VLM they used.

1) *PhyGenEval* for physical commnonsense alignment evaluation consists of three stages. We investigate the contribution of each stage to the final performance. Table 11 presents results using one or two stages (employing ensemble strategies

Table 10: Retrieval success rates for some different models.

| Model | Force (↑) | Light (↑) | Heat (↑) | Material (↑) | Overall (↑) |
|---|---|---|---|---|---|
| Cogvideo 5B | 0.7618 | 0.9013 | 0.8046 | 0.7815 | 0.8193 |
| Gen3 | 0.8353 | 0.9077 | 0.8627 | 0.8114 | 0.8577 |
| Pika | 0.7829 | 0.8777 | 0.7825 | 0.7736 | 0.8107 |
| Lavie | 0.7064 | 0.8328 | 0.7537 | 0.7219 | 0.7596 |
| Vchitect2 | 0.8078 | 0.9034 | 0.8317 | 0.7668 | 0.8327 |
| Keling | 0.8375 | 0.9018 | 0.8319 | 0.7978 | 0.8470 |
| Opensora | 0.8166 | 0.8755 | 0.8528 | 0.7707 | 0.8310 |
| Cogvideo 2B | 0.7924 | 0.8255 | 0.7886 | 0.7719 | 0.7971 |

when multiple VLMs are applicable). We find that optimal performance is achieved only when all three stages are used concurrently, demonstrating the rationale behind *PhyGenEval*'s design.

2) Given potential biases in single models and the costs associated with closed-source models, we offer two *PhyGenEval* computation methods: using GPT-4o or alternative open-source models (LLaVA-Interleave (Li et al., 2024) and InternVideo2 (Wang et al., 2024)). Table 12 shows that even using only small scale open-source models achieves a high correlation coefficient of 0.66. Notably, ensembling both methods yields the best results. Considering *PhyGenBench*'s relatively small size, we find this computational cost acceptable. Therefore we recommend users ensemble these methods.

3) We explore the performance of larger open-source models. Specifically, we replace GPT-4o used in the Physics Order Verification stage of *PhyGenEval* with InternVL2-Pro (78B), denoted as *PhyGenEval* (Open-L). Additionally, when we use smaller open-source model like llava-next-interleave 7B, we denote it as *PhyGenEval* (Open-S). We denote the original *PhyGenEval* with closed-source model like GPT-4o as *PhyGenEval* (Closed)

Results show that compared to smaller open-source models, the overall alignment coefficient with larger open-source models improves from 0.66 to 0.72, indicating that the method remains reproducible even when using exclusively open-source models. We believe that as open-source models continue to advance, they can achieve even better performance within *PhyGenEval* on *PhyGenBench*.

**The robustness of retrieval operations in *PhyGenEval*** In *PhyGenEval*'s Key Physical Phenomena Detection and Physics Order Verification, retrieval operations are required to detect key physical phenomena and their order in videos. Since retrieval may not always capture the target frames, we incorporate a regularization term $VLM(I_j, p_r)$ into the calculations of $S_{key}$ and $S_{order}$. VQAScore is used to evaluate the scores of retrieved frames and retrieval prompts, improving the robustness of retrieval operations. (Note: when the retrieval prompt is "Last Frame" or "Middle Frame," the default value is set to 1.)

Additionally, since *PhyGenBench* is intentionally designed with simple and common scenarios, retrieval operations achieve a relatively high success rate. We report the average $VLM(I_j, P_r)$ scores for different models in Table 10, showing that even Lavie achieves a retrieval score above 0.75, indicating generally high retrieval accuracy.

**The Component in *PhyGenEval* on semantic alignment evaluation.** we also perform necessary ablation experiments to validate the necessity of our SA evaluation design. Specifically, we compare: 1) VLM Model Selection: We leverage GPT-4o (Achiam et al., 2023) as a more robust VLM model for SA evaluation. 2) Effectiveness of our two-stage evaluation method proposed in Appendix C.1

1) As shown in Table 8, using GPT-4o in *PhyGenEval* is much better than using LLaVA, which achieve a higher Kendall's $\tau$ of 0.53 compared to 0.42, and a higher Spearman's $\rho$ of 0.56 versus 0.44. This indicates a stronger alignment between GPT-4o's evaluations and human annotations compared to open-source vlm models like Grid-LLaVA (Sun et al., 2024), justifying its selection as the preferred VLM model in the SA evaluation design. Since *PhyGenBench* includes a limited number of prompts, we believe that the cost of using GPT-4o is acceptable relative to the improvement in performance.

2) To validate the effectiveness of the two-stage strategy, we compare it with the method in T2V-CompBench (Sun et al., 2024), which directly uses Grid-LLaVA to apply the same scoring standard prompt for semantic alignment evaluation across all videos. For fairness, we also use Grid-LLaVA but implement the two-stage strategy proposed in Appendix C.1. As shown

Table 11: Comparison of PCA correlation results using each stage in *PhyGenEval*

| Metric | Mechanics | | Optics | | Thermal | | Material | | Overall | |
|---|---|---|---|---|---|---|---|---|---|---|
| | $\tau(\uparrow)$ | $\rho(\uparrow)$ | $\tau(\uparrow)$ | $\rho(\uparrow)$ | $\tau(\uparrow)$ | $\rho(\uparrow)$ | $\tau(\uparrow)$ | $\rho(\uparrow)$ | $\tau(\uparrow)$ | $\rho(\uparrow)$ |
| *PhyGenEval*-S | 0.50 | 0.54 | 0.43 | 0.45 | 0.50 | 0.54 | 0.72 | 0.77 | 0.56 | 0.61 |
| *PhyGenEval*-M | 0.46 | 0.49 | 0.49 | 0.53 | 0.55 | 0.59 | 0.53 | 0.57 | 0.55 | 0.60 |
| *PhyGenEval*-V | 0.26 | 0.30 | 0.44 | 0.47 | 0.33 | 0.35 | 0.48 | 0.52 | 0.42 | 0.46 |
| *PhyGenEval*-SM | 0.58 | 0.61 | 0.47 | 0.50 | 0.58 | 0.62 | 0.66 | 0.70 | 0.60 | 0.64 |
| *PhyGenEval*-SV | 0.56 | 0.59 | 0.41 | 0.43 | 0.58 | 0.60 | 0.70 | 0.74 | 0.59 | 0.62 |
| *PhyGenEval*-MV | 0.50 | 0.53 | 0.50 | 0.53 | 0.53 | 0.57 | 0.60 | 0.64 | 0.57 | 0.61 |
| *PhyGenEval* | **0.72** | **0.75** | **0.76** | **0.77** | **0.73** | **0.75** | **0.81** | **0.84** | **0.78** | **0.81** |

Table 12: Comparison of PCA correlation results using different models such as GPT-4o or open-sourced models in *PhyGenEval*

| Metric | Mechanics | | Optics | | Thermal | | Material | | Overall | |
|---|---|---|---|---|---|---|---|---|---|---|
| | $\tau(\uparrow)$ | $\rho(\uparrow)$ | $\tau(\uparrow)$ | $\rho(\uparrow)$ | $\tau(\uparrow)$ | $\rho(\uparrow)$ | $\tau(\uparrow)$ | $\rho(\uparrow)$ | $\tau(\uparrow)$ | $\rho(\uparrow)$ |
| *PhyGenEval* (Open-S) | 0.54 | 0.57 | 0.59 | 0.62 | 0.55 | 0.58 | 0.65 | 0.69 | 0.62 | 0.66 |
| *PhyGenEval* (Open-L) | *0.58* | *0.62* | *0.63* | *0.65* | *0.62* | *0.64* | *0.70* | *0.73* | *0.67* | *0.72* |
| *PhyGenEval* (Closed) | **0.72** | **0.75** | **0.76** | **0.77** | **0.73** | **0.75** | **0.81** | **0.84** | **0.78** | **0.81** |

in Table 8, *PhyGenEval*-Grid-LLaVA outperforms Grid-LLaVA, achieving a higher Kendall's $\tau$ score of 0.42 compared to 0.35, and a higher Spearman's $\rho$ score of 0.44 versus 0.39. This result demonstrates the effectiveness of our Two Stage Evaluation Method. By decomposing the evaluation into object detection and action detection, we effectively reduces the complexity of the task for VLMs in evaluating the sementic correctness of videos.

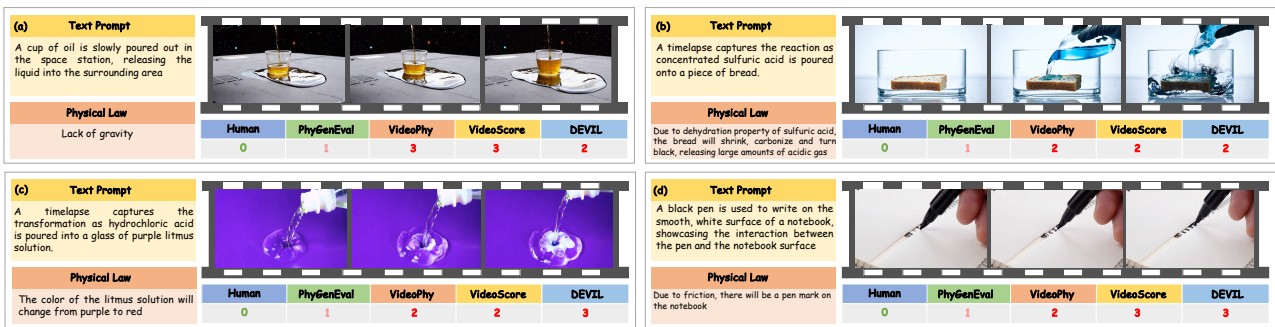

Figure 7: Visualization of some PhyGenEval error cases.

**Use real videos as reference** We agreed that using real videos as references might have provided more reference information. However, considering several difficulties:

1. It is challenging to collect real videos for each prompt.

2. Physical processes are diverse, making it difficult to collect unique real videos.

3. Due to differences in frame rates and other factors, video-to-video comparison is also challenging and might require training separate models.

Here, we adopted an alternative approach to incorporate real videos into the PhyGenEval framework.

Specifically, we extracted fifty Video-Caption pairs from WISA (Wang et al., 2025), belonging to mechanics, optics, and thermodynamics categories (WISA did not include physical property categories). We parsed the corresponding video captions into prompt and question formats as in PhyGenBench, and used PhysGenEval for evaluation. The results showed that real videos achieved extremely high scores under PhyGenEval, demonstrating the robustness of the framework.

|  | Mechanics (17 samples) | Optics (17) | Thermal (16) |
|---|---|---|---|
| Score | 0.93 | 0.95 | 0.93 |

We also tested the performance of open-source models on these 50 prompts, using machine scores and machine scores / machine scores of real videos (the latter serving as reference scores after error elimination), obtaining the following table:

|  | Mechanics | Optics | Thermal | Avg |
|---|---|---|---|---|
| CogVideoX2B | 0.36(0.37) | 0.44(0.49) | 0.33(0.41) | 0.39(0.39) |
| CogVideoX5B | 0.36(0.36) | 0.52(0.57) | 0.47(0.53) | 0.46(0.50) |
| Opensora V1.2 | 0.36(0.38) | 0.49(0.52) | 0.39(0.39) | 0.43(0.45) |
| Lavie | 0.23(0.27) | 0.42(0.45) | 0.36(0.40) | 0.36(0.38) |
| Vchitect 2.0 | 0.41(0.43) | 0.52(0.57) | 0.42(0.44) | 0.47(0.50) |
| Hunyuan | 0.46(0.49) | 0.53(0.55) | 0.39(0.40) | 0.48(0.51) |
| Pyramid Flow (Flux) | 0.33(0.37) | 0.50(0.54) | 0.44(0.50) | 0.44(0.48) |
| Pyramid Flow (Sd3) | 0.43(0.54) | 0.46(0.52) | 0.33(0.40) | 0.42(0.49) |

The Spearman coefficient of model rankings calculated by these two methods is **0.90**, indicating that the current evaluation framework can achieve robust results.

# E. Discussion

**Error case analysis.** As shown in Figure 7, we visualize some error cases where both PhyGenEval and competing methods like DEVIL fail to correctly identify the physical realism of the videos. These error cases are often caused by ***confusing but iconic physical phenomena*** in the videos that do not align with the correct progression of physical processes (e.g., in the erroneous case of the "burnt bread" experiment, black coloration appears but does not align with the expected phenomenon), leading to misjudgments. However, even in these cases, PhyGenEval remains closer to human ratings compared to other methods.

**The Impact of Scaling on Physical Commonsense in Video Generation.** Scaling laws have been extensively validated in video generation models (Kaplan et al., 2020). We investigate their efficacy in addressing the challenges of physical commonsense presented in *PhyGenBench*. As shown in Table 2, CogVideo 5B demonstrates improvements over CogVideo 2B, albeit with limited progress in the Mechanics category. Our qualitative analysis, illustrated in Figure 8, reveals significant advancements in static scenes with CogVideo 5B. It accurately captures complex phenomena such as colorful bubbles resulting from interference and diffraction, and oxidation-induced rusting of iron. In thermal, despite imperfections, CogVideo 5B generates more realistic boiling simulations compared to its predecessor. However, both models struggle with simple motion dynamics, exemplified by their inability to accurately depict a bouncing football. We posit that while scaling up enhances the model's capacity to generate videos that align with physical commonsense for individual objects, it may be insufficient for physical phenomenons involving dynamic physical laws. Addressing these challenges likely requires extensive training on carefully curated synthetic data, as suggested by (Liu et al., 2024b). This approach could potentially bridge the gap in the model's grasp of fundamental physical laws.

**The robustness of low-quality videos.** Although we deliberately select simple scenarios when constructing *PhyGenBench* to make it easier for models to generate videos that align with prompts while exposing physical errors, certain T2V models

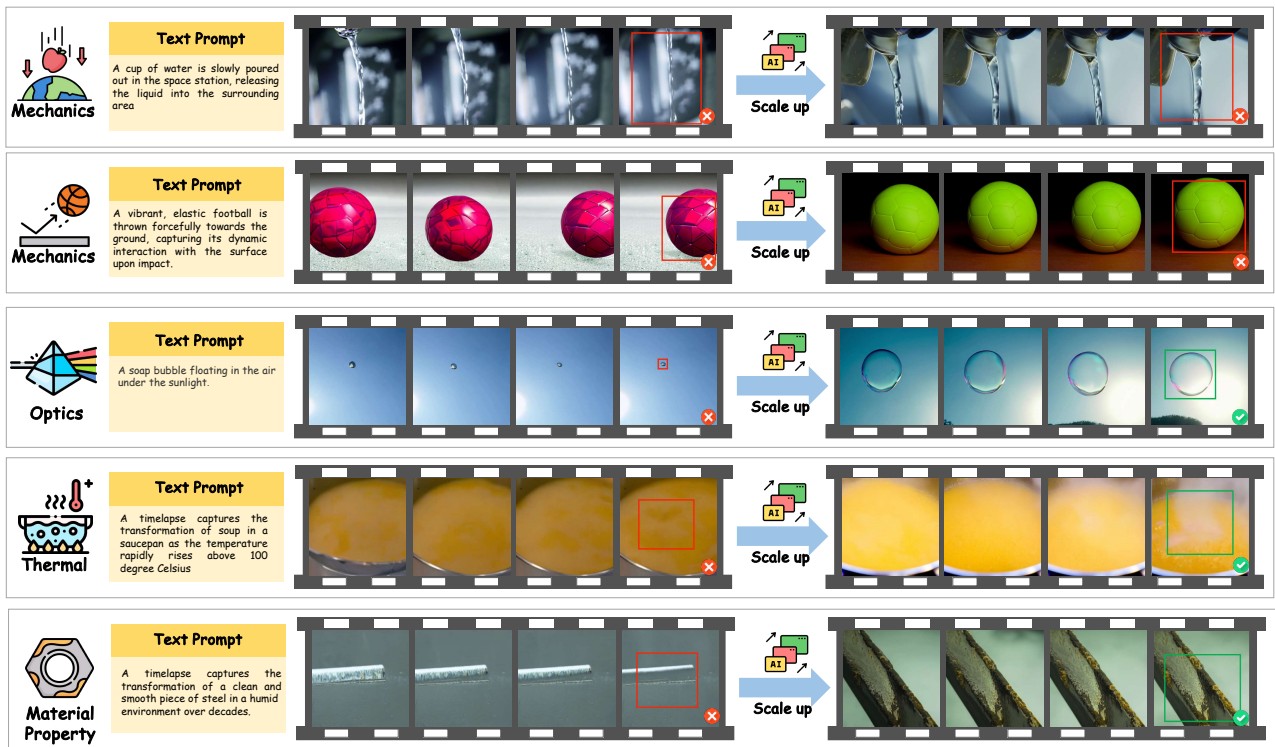

Figure 8: The qualitative comparison of CogVideoX 2B and CogVideoX 5B. The result shows that simply scaling up can solve some issues, but dynamic physical phenomenons involving the design of motion patterns remain challenging.

may still perform poorly on specific prompts. In this regard, *PhyGenEval* demonstrates a degree of robustness. We select 100 low-quality videos and calculated their average PCA Score, which is $0.14$, significantly lower than the average PCA Score in Table 2. Figure 9 provides visualizations of some low-quality videos, most of which fail to meet the prompt requirements, resulting in very low physical scores.

**The effect of fine-tuningect of fine-tuning.** WISA fine-tunes CogVideo using real text-video pairs and tests it on PhyGenBench. The results show that the score increases from 0.41 to 0.43. Although there is some improvement, it is not significant. Therefore, it is still necessary to design fine-tuning methods that explicitly incorporate physical laws, or to use more powerful video generation model backbones.

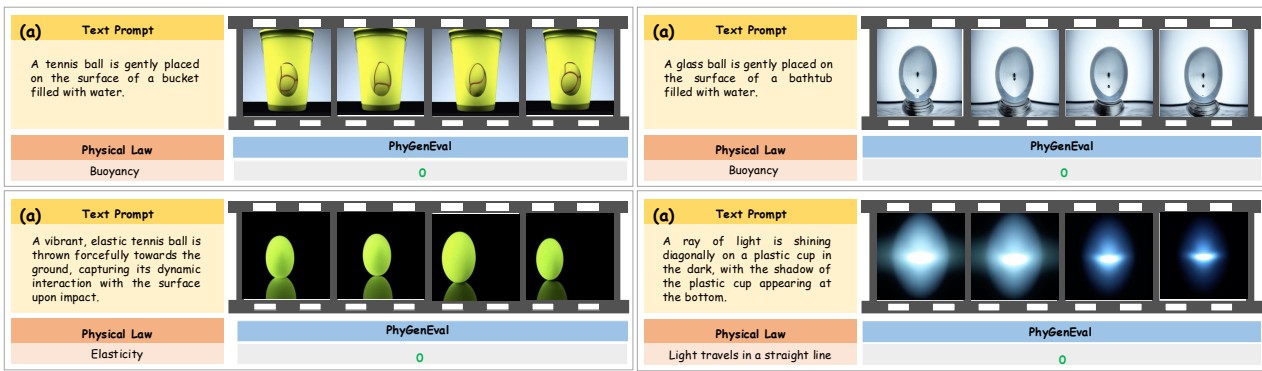

Figure 9: Visualization of some low quality videos and their PCA score.

Table 13: Resource consumption of models used in PhyGenEval.

| Model | Batch Size | Resources | Times | Memory Utilization Per GPU |
|---|---|---|---|---|
| GPT-4o(stage2) | 8 | USD 1.4 | 5min | - |
| GPT-4o(stage3) | 8 | USD 3.1 | 5min | - |
| LLaVA-Next-Interleave-7B | 1 | 1 x A100-80GB | 2min | 20408MiB |
| VQAScore | 3 | 1 x A100-80GB | 10min | 72726MiB |
| InternVideo | 1 | 1 x A100-80GB | 1min | 7766MiB |

# F. Computaional Resources

Although our method involves different stages, it remains straightforward. Table 13 provides a summary of the resource consumption, showing that the entire evaluation process can be completed quickly and at low cost.

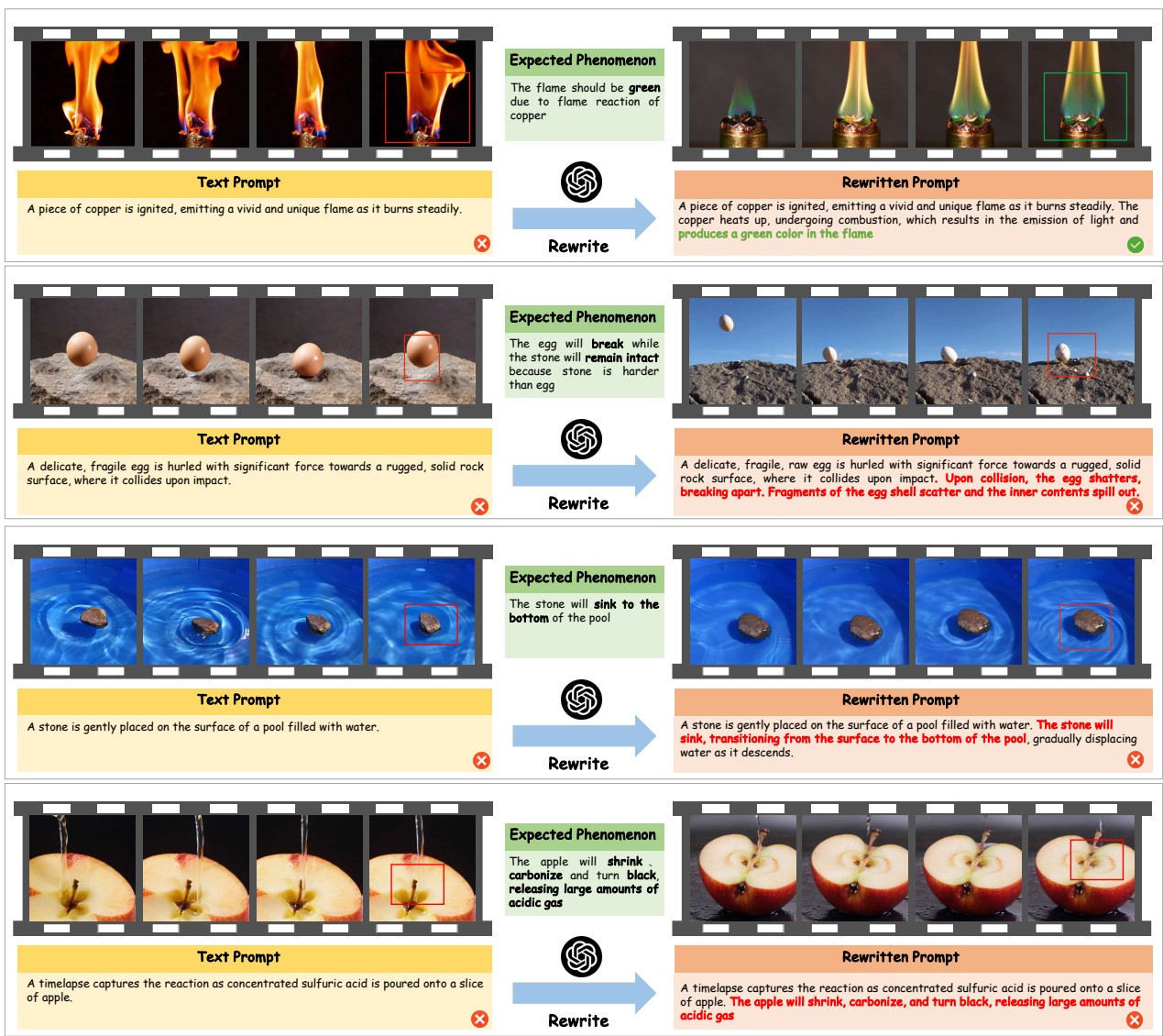

Figure 10: The qualitative comparison of effects before and after using rewritten prompts. The results indicate that rewriting prompts addresses only a few basic issues (such as flame color reactions), while the majority of problems remain unsolved.

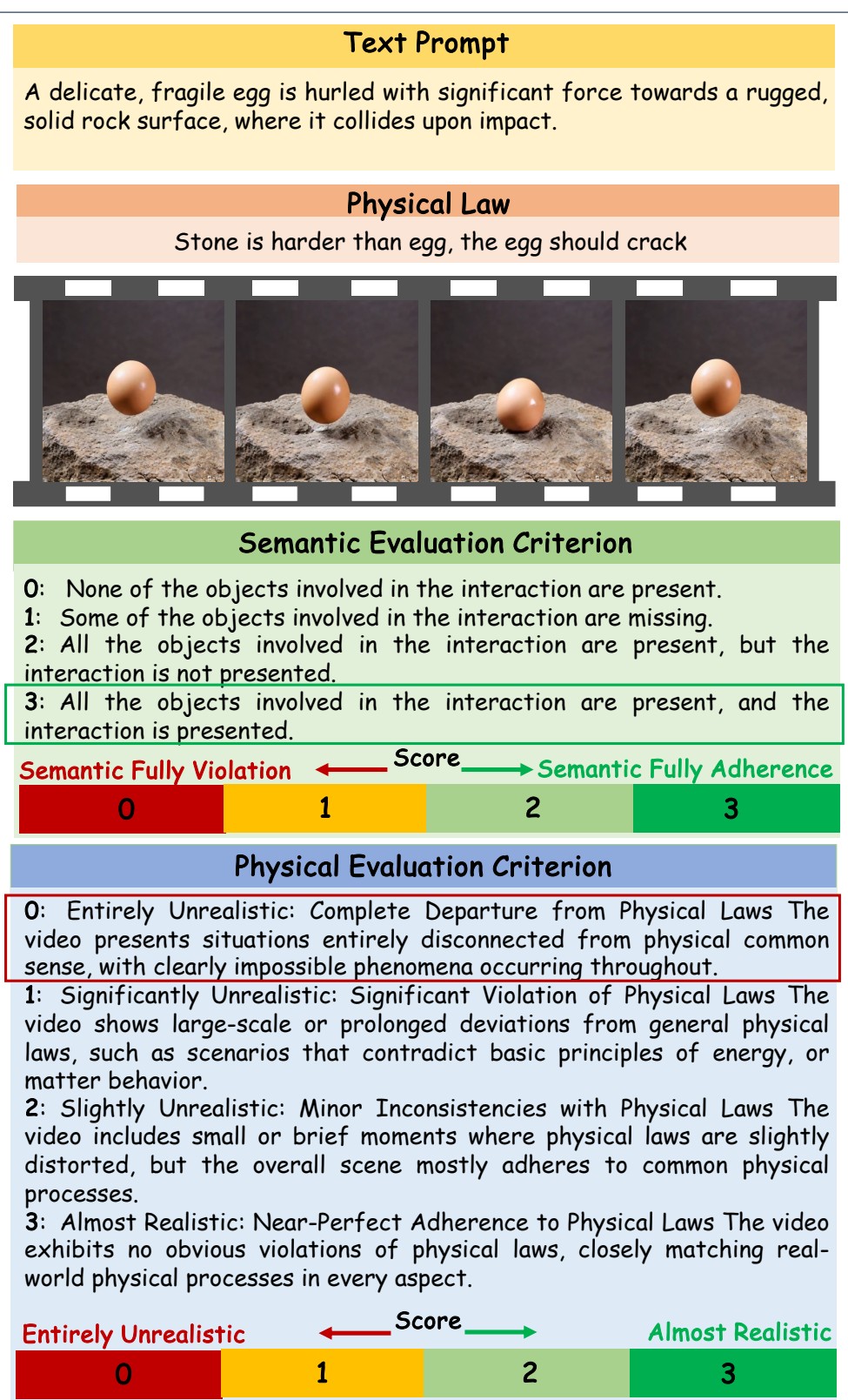

Figure 11: Detailed diagram of the human evaluation process. We ask the annotators to score the semantic alignment and physical commonsense alignment of the video according to the scoring criteria in the figure.

