# OpenReview forum: "Towards World Simulator: Crafting Physical Commonsense-Based Benchmark for Video Generation"
_ICML.cc/2025/Conference — ICML 2025 poster_

### Official Review · Reviewer_uagC · 2025-03-11

**Overall Recommendation:** 3

**Summary:**

The paper introduces PhyGenBench, a benchmark designed to evaluate whether Text-to-Video models accurately adhere to fundamental physical laws. The study aims to assess how well these models can simulate intuitive physics, which is considered essential for developing a general world simulator. To systematically evaluate T2V models, the paper proposes a hierarchical evaluation method, PhyGenEval, which assesses the physical commonsense correctness of generated videos. The study evaluates state-of-the-art T2V models.

**Claims And Evidence:**

The claims made in the submission are generally supported by clear and well-structured evidence, but some areas may require further validation or refinement.

The use of Vision-Language Models and GPT-4o can introduce potential biases. The paper does not provide a robust error analysis of PhyGenEval’s failure cases.

The paper positions PhyGenBench as a step toward general world simulation. However, no discussion is provided on how improving physics modeling would integrate into broader world simulation goals. How does this work compare to embodied AI efforts in world modeling?

**Essential References Not Discussed:**

The paper claims that current T2V models fail at intuitive physics but does not cite prior work on physics-based video generation, where researchers have attempted to incorporate physical constraints into generative models.

PhysGen: Rigid-Body Physics-Grounded Image-to-Video Generation
InterDyn: Controllable Interactive Dynamics with Video Diffusion Models

**Experimental Designs Or Analyses:**

Yes, I analyzed the soundness and validity of the experimental design and analysis in the paper. Overall, the experiments are well-structured and provide valuable insights.

The paper’s three-level evaluation framework (PhyGenEval) effectively assesses physical commonsense in T2V models. By breaking down evaluation into Key Physical Phenomena Detection, Physics Order Verification, and Overall Naturalness, it ensures a systematic, structured analysis aligned with human judgment.

The wide-ranging model evaluation ensures fair comparisons and highlights that even top models struggle with intuitive physics, reinforcing the gap between AI-generated videos and real-world simulation.

Limitations:

The study evaluates T2V models only against text-based descriptions, rather than comparing outputs with real-world physics simulations. This could limit the reliability of the benchmark, as models may generate visually plausible but physically incorrect videos that still align with textual prompts. Incorporating physics simulation datasets as a reference could provide a more objective evaluation of physical correctness in generated videos.

Some physical laws require more context than what is provided in the text-based prompts. For example, a prompt about an object floating or sinking in water may not specify material density, leading to ambiguous interpretations. Ensuring that prompts contain clear and complete physical conditions would improve the accuracy of assessments.

A key concern is that PhyGenEval relies on GPT-4o and Vision-Language Models for scoring. If GPT-4o has flaws in its understanding of physics, it could introduce biases into the evaluation. The study does not analyze cases where PhyGenEval disagrees with human evaluators, making it difficult to assess its true reliability. Conducting an error analysis could help identify potential weaknesses in the evaluation framework.

Additionally, while the study claims that scaling models alone does not improve physical reasoning, it does not test whether fine-tuning on physics-specific datasets could lead to significant improvements.

**Methods And Evaluation Criteria:**

Yes, the proposed methods and evaluation criteria, including PhyGenBench and PhyGenEval, generally make sense for evaluating the physical commonsense capabilities of T2V models.

The use of GPT-4o and other VLMs raises concerns about potential biases and failure cases (e.g., if the VLM itself has poor physical commonsense).

The paper argues that scaling models alone does not solve physical reasoning issues, but does not test whether fine-tuning on physics-specific datasets could help.

The paper positions PhyGenBench as a step toward general world simulation, but does not explicitly compare its approach to other physics-focused AI evaluations (e.g., embodied AI or reinforcement learning environments).

**Other Comments Or Suggestions:**

Typos:

L1075: "we effectively reduces" -> "we effectively reduce"

"sementic" --> "semantic"

**Other Strengths And Weaknesses:**

Strengths:

One of the paper's key strengths is its originality in addressing a critical gap in T2V evaluation. While prior research has focused on visual quality, motion coherence, and spatial relationships, this study is one of the first to systematically assess physical commonsense adherence in generated videos.

The three-tier evaluation structure in PhyGenEval is another significant contribution. Unlike traditional metrics like Fréchet Video Distance (FVD) and VideoScore, which primarily measure perceptual fidelity, the proposed framework evaluates physical commonsense in the generated videos.

Weaknesses:

PhyGenBench evaluates T2V models only against text-based descriptions, rather than comparing them to real-world physics simulations. This raises concerns about whether the benchmark truly reflects real-world physics fidelity.

While the study argues that PhyGenEval aligns well with human judgment, it does not analyze failure cases where GPT-4o misinterprets physics. Since LLMs and VLMs may not have a strong grasp of causality or dynamics, their scoring could introduce systematic biases.

**Questions For Authors:**

The paper evaluates T2V models against text-based descriptions rather than real-world physics simulations or recorded videos. Have you considered comparing generated videos with real physics-based datasets? If not, how do you justify this omission, given that real-world comparisons would provide a more objective evaluation?

The evaluation framework relies on GPT-4o and Vision-Language Models. How do you ensure that these models correctly assess physical realism, given that VLMs are not explicitly trained for physics verification?

Some physical laws require additional context that may be missing from text-based prompts. How do you ensure that prompts are unambiguous and do not introduce interpretation biases?

How well do you expect PhyGenBench to generalize as future T2V models improve? Will the benchmark need updating as models become more sophisticated, or is it designed to remain relevant long-term?

**Relation To Broader Scientific Literature:**

The paper builds on advancements in T2V generation, which has rapidly improved in terms of visual quality, motion coherence, and scene complexity. Models like Sora, Gen-3, and Pika can generate high-resolution videos from text prompts, but they lack an understanding of physical commonsense. Prior research in video generation has largely focused on aesthetic quality and temporal consistency, rather than ensuring that generated videos follow real-world physics. This paper addresses that gap by introducing PhyGenBench, a benchmark that explicitly evaluates whether T2V models generate videos that align with fundamental physical laws.

Traditional metrics like Fréchet Video Distance (FVD) assess video quality but fail to measure physical correctness. The paper introduces PhyGenEval, a three-tier evaluation framework that combines Vision-Language Models and GPT-4o to assess physical correctness in generated videos. This approach moves beyond simple perceptual metrics and provides a structured way to measure how well AI models understand physics.

**Theoretical Claims:**

The paper primarily focuses on benchmarking and evaluation of T2V models rather than presenting formal theoretical claims with rigorous proofs.

---

> ### Author Rebuttal · Authors · 2025-04-01
>
> We appreciate your suggestions, which are essential for enhancing the paper. We address all questions sequentially and will incorporate those details in the revisions.
>
> Q1:Comparing generated videos with real physics to eval.
>
> A1:We considered using real videos as references to eval but encountered some scalability challenges. Collecting gt videos for each prompt was difficult, as physical processes varied greatly across scenarios, making a single deterministic video insufficient as a golden reference. Additionally, evaluating semantic consistency across videos with different frame rates and scenarios remained an unresolved challenge.
>
> Instead, we incorporated real world physical videos into PhyGenEval as references. We sampled 50 real videos along with detailed captions from WISA[1]. After parsing these captions into the PhyGenBench format and evaluting the real videos, our evaluation showed they achieved extremely high physical alignment scores under PhyGenEval, serving a strong reference baselines for other models to compare. And demonstrated the robustness of PhyGenEval for real physical scenarios.
>
> | Mechanics(17 samples) | Optics(17) | Thermal(16) |
> | -- | -- | -- |
> | 0.93 | 0.95 | 0.93 |
>
> We further tested the performance of 8 models on these prompts, using real physical video scores as a normalization score to for more realistic comparison (e.g., model generated videos score / real physical videos score).
>
> |   | Mechanics | Optics |Thermal | Avg. |
> | -- | -- | -- | -- | -- |
> | CogVideoX2B | 0.36\0.37 | 0.44\0.49 | 0.33\0.41 | 0.39\0.39 |
> | CogVideoX5B  | 0.36\0.36 | 0.52\0.57 | 0.47\0.53 | 0.46\0.50 |
> | OpensoraV1.2 | 0.36\0.38 | 0.49\0.52 | 0.39\0.39 | 0.43\0.45 |
> | Lavie  | 0.23\0.27 | 0.42\0.45 | 0.36\0.40 | 0.36\0.38 |
> | Vchitect2.0 | 0.41\0.43 | 0.52\0.57 | 0.42\0.44 | 0.47\0.50 |
> | Hunyuan | 0.46\0.49 | 0.53\0.55 | 0.39\0.40 | 0.48\0.51 |
> | Pyramid Flow(flux) | 0.33\0.37 | 0.50\0.54 | 0.44\0.50 | 0.44\0.48 |
> | Pyramid Flow(sd3) | 0.43\0.54 | 0.46\0.52 | 0.33\0.40 | 0.42\0.49 |
>
> The Spearman of model rankings calculated by these two methods was **0.90**, indicating the robustness of PhyGenEval. And this could be further improved by incorporating real physical video scores. We will include this in the revision.
>
> Q2:How do you ensure that these models correctly assess physical realism.
>
> A2:Compared with VLMs, LLMs were trained on real physics reasoning datasets, demonstrating better physics understanding capabilities. Thus, we addressed VLM evaluation limitations by using LLMs to understand physical laws during PhyGenBench construction, reducing physics comprehension difficulty. In PhyGenEval, we also incorporated priors from physical laws and evaluated basic principles step by step(e.g.,The egg breaks after it hits the stone), lowering the evaluation difficulty. As demonstrated in Table 10, PhyGenEval significantly exceeded the direct evaluation capabilities of GPT-4o.
>
> Q3:Some physical laws require additional context.
>
> A3:As shown in Figure 2b, we addressed exactly this issue through a prompt augmentation stage. For example, we transformed "egg collides stone" into "fragile egg was hurled with significant force towards solid rock," to eliminated potential ambiguities. Furthermore, Quality Control check for prompt completeness also verified it. (line 209).
>
> Q4:Expect PhyGenBench to generalize.
>
> A4:Currently, we focused on basic physical laws that effectively revealed limitations in existing T2V models. As simulation engines and T2V models became more powerful, we hoped to generate real reference videos for each prompt in the future, and trained video-to-video scoring models through them.
>
> Q5:FT on physics-specific dataset.
>
> A5:We randomly selected 1200 Video-Text pairs from WISA and performed lora(r=128) fine-tuning on CogVideoX 5B. The results indicated that this did not solve the problem. We believed this might due to: The training set was too small to cover enough domains; The base model was not strong enough, making it difficult to generalize; More explicit injection of physical laws was needed, such as training with synthetic videos.
>
> | Model | Mech. | Opt. | The. | Mat. | Avg.  |
> | -- | -- | -- | -- | -- | -- |
> | CogVideoX 5B | 0.39  | 0.55 | 0.40 | 0.42 | 0.449 |
> | +FT | 0.38  | 0.58 | 0.41 | 0.40 | 0.453 |
>
> Q6:Error analysis.
>
> A6:We provided some error cases in line 1094. Besides, We collected 50 videos where machine scores differ from human. The statistical information is:
>
> |Type | Per. | Avg Diff. (0-3)|
> | -- | -- | -- |
> | Higher | 90%  | 1 |
> | Lower  | 10%  | 1  |
>
> We defined 3 core error cases: spatial, semantic, and temporal understanding errors. The results show that most are due to temporal understanding, which we will improve it in the future.
>
> | Sem. | Spa. | Tem. |
> | ---- | ---- | ---- |
> | 28%| 12%| 64% |
>
> Others:We have already cited PhysGen(line 1112).We will update InterDyn in the revision.
>
> [1] WISA: World Simulator Assistant for Physics-Aware Text-to-Video Generation

---

### Official Review · Reviewer_AK75 · 2025-03-11

**Overall Recommendation:** 3

**Summary:**

The paper introduces a benchmark designed to assess the extent to which generative video models internalize physical laws. The authors construct a dataset comprising 160 prompts that incorporate 27 physical phenomena. Additionally, they propose a method leveraging large vision-language models to automatically evaluate the physical correctness of the generated videos. They evaluate 8 open-source models and 6 proprietary models.

**Claims And Evidence:**

They claim that scaling and prompt engineering alone do not significantly improve physical commonsense.
They claim that there automated evaluation method is suited to evaluate physical commonsense, which is confirmed by the correlation with human evaluations and by the fact that the evaluation is robust to changes that affect visual quality without affecting the physical correctness.

**Essential References Not Discussed:**

N/A

**Experimental Designs Or Analyses:**

Each video was evaluated by 3 independent annotators.

**Methods And Evaluation Criteria:**

The authors create there own benchmark and evaluation method that seems robust to changes that affect visual quality without affecting the physical correctness.

**Other Comments Or Suggestions:**

N/A

**Other Strengths And Weaknesses:**

Strength: The paper is clear. The score decomposition used in the automatized evaluation process makes a lot of sense.
Weakness: The evaluation is done using only 160 which might not be significant enough. Would it be possible to perform the evaluation on 1000 prompts and show that the score ordering does not change? Or alternatively, sample 10 sets of 100 prompts among the 160 and compute the variance. I am happy to raise my score if the authors can show that the computed score is low-variance.

**Questions For Authors:**

N/A

**Relation To Broader Scientific Literature:**

Evaluating the adherence of large models to physical laws is a very important direction, given the wide adoption of these models in the industry.

**Theoretical Claims:**

N/A

---

> ### Author Rebuttal · Authors · 2025-04-01
>
> Many thanks for your feedback, which is vital to improving our paper's standard. We respond to all inquiries in sequence and will incorporate those details in the revision.
>
> Q1: The evaluation is done using only 160 which might not be significant enough...
>
> A1: Following your suggestion, we randomly selected 100 prompts from the 160 prompts and repeated this 10 times. We calculated the Spearman coefficient and Kendall correlation coefficient between the model's ranking and the ranking using all 160 prompts. The results were shown in the table below, reflecting the similarity of the model rankings and indicated that a low variance existed within the current prompts set.
>
> | Spearman | Kendall |
> | -------- | ------- |
> | 0.87     | 0.82    |
>
> Our current 160 prompts focuses on the most basic physical laws and scenarios, which should be sufficient to expose problems in the models. As T2V models develop, we will gradually expand to include new physical laws and design more complex scenarios that reflect physical principles.

---

### Official Review · Reviewer_bpjp · 2025-03-14

**Overall Recommendation:** 4

**Summary:**

The paper introduces PhyGenBench, a benchmark specifically designed to assess text-to-video (T2V) models on their ability to generate physically plausible videos grounded in intuitive physics. It comprises 160 carefully constructed prompts covering 27 distinct physical laws across four fundamental domains: mechanics, optics, thermal phenomena, and material properties. Additionally, the authors propose PhyGenEval, a novel hierarchical evaluation framework leveraging vision-language models (VLMs) and GPT-4o to measure semantic alignment and physical commonsense alignment in generated videos. Their extensive evaluation reveals that existing T2V models significantly underperform in accurately generating physically correct phenomena, indicating substantial room for improvement toward genuine world simulators.

**Claims And Evidence:**

The claims made in the submission are clearly supported by convincing evidence. The authors provide comprehensive evaluations, demonstrating that current T2V models fail to capture intuitive physical laws robustly. Experimental results are thoroughly presented, including comparisons across various state-of-the-art models, clearly highlighting their shortcomings in physical correctness.

**Essential References Not Discussed:**

The authors have discussed relevant literature. However, explicitly referencing prior efforts in physics-based video synthesis or evaluation could strengthen the context provided.

**Experimental Designs Or Analyses:**

The experimental designs and analyses are sound. The authors clearly document their evaluation methodology, including human assessments, which validate their automated metric's high alignment with human judgments.

**Methods And Evaluation Criteria:**

The claims made in the submission are clearly supported by convincing evidence. The authors provide comprehensive evaluations, demonstrating that current T2V models fail to capture intuitive physical laws robustly. Experimental results are thoroughly presented, including comparisons across various state-of-the-art models, clearly highlighting their shortcomings in physical correctness.

**Other Comments Or Suggestions:**

NO

**Other Strengths And Weaknesses:**

NO

**Questions For Authors:**

Can the PhyGenEval framework be easily adapted to new physical phenomena, or does this require extensive manual recalibration?

Have you explored incorporating simulation-generated videos as references in your benchmark, and could this improve the robustness of your evaluations?

Would it be feasible to automatically generate prompts using generative models or reinforcement learning to improve scalability and diversity further?

**Relation To Broader Scientific Literature:**

This paper makes a good contribution by addressing a gap in current benchmarks, which primarily focus on visual quality or semantic alignment, by explicitly assessing intuitive physics in T2V models. The authors position their work effectively relative to existing benchmarks like VideoPhy, VideoScore, and DEVIL, highlighting the unique aspects and strengths of their proposed methods.

**Theoretical Claims:**

The paper does not primarily rely on theoretical proofs, so this section is not applicable.

---

> ### Author Rebuttal · Authors · 2025-04-01
>
> We're grateful for your suggestions, which plays a critical role in elevating our paper's quality. We tackled all questions one by one and will incorporate those details in the revision.
>
> Q1: Can the PhyGenEval framework be easily adapted to new physical phenomena
>
> A1: We sampled 50 prompts from WISA and applied 8 open-source models to generate videos. For each model, we sampled 20 videos and conduct human evaluation. We then calculated the human alignment coefficient with the model scores, resulting in the table below. As can be seen, PhyGenEval demonstrates certain generalization capabilities beyond the prompts in PhyGenBench.
>
> | Mechanics | Optics | Thermal | Avg. |
> | --------- | ------ | ------- | ---- |
> | 0.71      | 0.76   | 0.76    | 0.74 |
>
> Q2: Have you explored incorporating simulation-generated videos as references in your benchmark
>
> A2: We agreed that using real videos as references might have provided more reference information. However, considering several difficulties:  1. It is challenging to collect real videos for each prompt 2. Physical processes are diverse, making it difficult to collect unique real videos 3. Due to differences in frame rates and other factors, video-to-video comparison is also challenging and might require training separate models. Here, we adopted an alternative approach to incorporate real videos into the PhyGenEval framework.
>
> Specifically, we extracted fifty Video-Caption pairs from WISA, belonging to mechanics, optics, and thermodynamics categories (WISA did not include physical property categories). We parsed the corresponding video captions into prompt and question formats as in PhyGenBench, and used PhysGenEval for evaluation. The results showed that real videos achieved extremely high scores under PhyGenEval, demonstrating the robustness of the framework.
>
> | Mechanics(17 samples) | Optics(17) | Thermal(16) |
> | --------------------- | ---------- | ----------- |
> | 0.93                  | 0.95       | 0.93        |
>
> We also tested the performance of open-source models on these 50 prompts, using machine scores and machine scores / machine scores of real videos (the latter serving as reference scores after error elimination), obtaining the following table:
>
> |                      | Mechanics  | Optics     | Thermal    | Avg        |
> | -------------------- | ---------- | ---------- | ---------- | ---------- |
> | CogVideoX2B          | 0.36(0.37) | 0.44(0.49) | 0.33(0.41) | 0.39(0.39) |
> | CogVideoX5B          | 0.36(0.36) | 0.52(0.57) | 0.47(0.53) | 0.46(0.50) |
> | Opensora V1.2        | 0.36(0.38) | 0.49(0.52) | 0.39(0.39) | 0.43(0.45) |
> | Lavie                | 0.23(0.27) | 0.42(0.45) | 0.36(0.40) | 0.36(0.38) |
> | Vchitect 2.0         | 0.41(0.43) | 0.52(0.57) | 0.42(0.44) | 0.47(0.50) |
> | Hunyuan              | 0.46(0.49) | 0.53(0.55) | 0.39(0.40) | 0.48(0.51) |
> | Pyramid Flow（Flux） | 0.33(0.37) | 0.50(0.54) | 0.44(0.50) | 0.44(0.48) |
> | Pyramid Flow（Sd3）  | 0.43(0.54) | 0.46(0.52) | 0.33(0.40) | 0.42(0.49) |
>
> The Spearman coefficient of model rankings calculated by these two methods is **0.90**, indicating that the current evaluation framework can achieve robust results.
>
> Q3: Would it be feasible to automatically generate prompts using generative models or RL
>
> A3: 1. We currently used LLMs (e.g., GPT-4o) in the Diverse Enhancement and question generation steps of prompt construction to expand prompts and parse physical laws. Afterward, we conducted detailed manual reviews to control the quality of PhyGenBench. In the future, we will further explore automated approaches to generate high-quality prompts, reducing human effort in this process.
>
> 2.We will consider using the automated prompt construction method mentioned above to generate a training set, then use various T2V models to generate videos and conduct human scoring to provide reward signals. Subsequently, we can train T2V models using processes like DPO. Alternatively, we can use human scoring to train reward models and implement RLHF algorithms like PPO to optimize the models' physical understanding capabilities.
>
> Q4: physics-based video synthesis or evaluation could strengthen the context provided...
>
> A4: We will discuss these directions you mentioned in the related work section of the next version of our paper. For example, PhysGen[1] simulating videos of rigid body motion; PhysMotion[2] simulating real I2V scenarios, etc.
>
> Your suggestions are greatly valued, and please let us know if you need any clarification.
>
> [1] PhysGen: Rigid-Body Physics-Grounded Image-to-Video Generation
>
> [2] PhysMotion: Physics-Grounded Dynamics From a Single Image

---

### Official Review · Reviewer_QPju · 2025-03-19

**Overall Recommendation:** 4

**Summary:**

This paper introduces PhyGenBench, a benchmark assessing physical commonsense correctness in Text-to-Video (T2V) models, and PhyGenEval, an automated evaluation framework. PhyGenBench includes 160 prompts covering 27 physical laws across mechanics, optics, thermal, and material properties, ensuring a comprehensive assessment. PhyGenEval evaluates key physical phenomena, causal order, and overall naturalness using Vision-Language Models (VLMs) and Large Language Models (LLMs). The study shows that scaling models and prompt engineering alone are insufficient, emphasizing the need for better physics-aware video generation. PhyGenBench and PhyGenEval provide a scalable and structured evaluation framework, encouraging advancements in realistic world simulation.


## update after rebuttal
This is an important and timely contribution to the community. The authors propose a benchmark and evaluation methods for video generation. I am inclined to recommend acceptance of the work.

**Claims And Evidence:**

Most of claims are supported by author's evidence.
The author conducted experiments to support the core findings: 1. current T2V models struggle with physical commonsense is well-supported by experimental results from 14 models, with Gen-3 achieving only 0.51 in physical commonsense accuracy. 2. The hierarchical evaluation strategy in PhyGenEval such as Key Physical Phenomena Detection, Physics Order Verification, and Overall Naturalness is clearly explained and validated through correlation with human ratings. This provides convincing evidence that it is a more effective metric than existing alternatives like VideoScore and VideoPhy.
However, the claim that scaling models and prompt engineering alone cannot resolve physical commonsense issues is partially supported, more ablation studies on different training techniques (e.g., incorporating physics-based priors) could provide deeper insights.

**Essential References Not Discussed:**

One relevant technical report that assesses the physical understanding of video generation models and could be discussed is "How Far is Video Generation from World Model? – A Physical Law Perspective." This work examines the extent to which T2V models adhere to fundamental physical laws, providing additional context for evaluating physical commonsense in generative models.

**Experimental Designs Or Analyses:**

The experiment designs and analyses are sufficent.

**Methods And Evaluation Criteria:**

The proposed method and criteria make sense to me. The benchmark PhyGenBench overs 160 prompts across 27 physical laws, ensuring a diverse and structured evaluation of mechanics, optics, thermal, and material properties. It contains three-tiered evaluation: key physical phenomena detection, physics order verification, and overall naturalness, breaks down physical correctness into measurable components. Besides, the performance aligns with human evaluations.
While physics order verification evaluates causality, there is no explicit check for temporal smoothness in video sequences (e.g., abrupt frame transitions violating motion continuity). Additionally, it primarily relies on key frame detection using CLIPScore. However, the accuracy of CLIPScore in reliably identifying key frames and detecting physical phenomena remains unclear, warranting further validation.

**Other Comments Or Suggestions:**

My comments can be found in the above 'Other Strengths And Weaknesses' section.

**Other Strengths And Weaknesses:**

Overall, I think this is an important work for the community. My comments about the strengths and weaknesses can be seen as follows:
Strengths:
1. The paper addresses a crucial gap in text-to-video (T2V) generation by introducing a benchmark that evaluates physical commonsense, an aspect largely overlooked in existing works. The combination of a structured benchmark (PhyGenBench) and a hierarchical evaluation framework (PhyGenEval) provides a novel and scalable approach to assessing physical correctness in generative models.
2. The evaluation includes 14 T2V models, comparing their physical commonsense accuracy (PCA) scores, and correlates automated assessments with human evaluations.
3.By emphasizing intuitive physics in generative AI, the work could influence future developments in video synthesis, robotics simulation, and AI-driven scientific visualization, expanding the application of T2V models beyond entertainment.

Weaknesses:
1. While PhyGenBench covers a diverse range of physical laws, its applicability to unseen, real-world scenarios remains unclear. Further validation on dynamically generated prompts or real-world physics-based tasks would improve its robustness.
2. The evaluation method depends on CLIPScore to locate key frames, but its accuracy in reliably detecting physical phenomena is not well-validated, which may introduce errors in assessment.
3. While physics order verification ensures correct event sequences, the framework does not explicitly assess motion coherence, which is critical for realistic video generation. Integrating temporal smoothness metrics could strengthen the evaluation.

**Questions For Authors:**

My questions can be found in the above 'Other Strengths And Weaknesses' section.

**Relation To Broader Scientific Literature:**

The paper builds upon prior research in text-to-video (T2V) generation, physical commonsense reasoning, and evaluation benchmarks for generative models, while addressing critical gaps in these areas. Previous benchmarks for text-to-video (T2V) models (e.g., VBench, EvalCrafter) primarily evaluate motion smoothness, spatial consistency, and overall video quality, but they do not assess physical correctness, a gap that PhyGenBench aims to fill. While prior works like Physion and ContPhy focus on physical reasoning in vision-language models for prediction tasks, they do not evaluate generative capabilities, whereas PhyGenBench assesses whether T2V models can generate physically plausible videos rather than just recognizing or predicting physical events. Additionally, existing VLM-based evaluation methods (e.g., VideoScore, VideoPhy) struggle to detect violations of physical laws, but PhyGenEval introduces a hierarchical evaluation framework that explicitly verifies key physical phenomena, causal order, and overall naturalness, making it more aligned with real-world physics principles. By introducing PhyGenBench and PhyGenEval, the paper advances the scientific understanding of physics-aware video generation, providing a scalable and automated method for evaluating physical commonsense in generative models.

**Theoretical Claims:**

There is no theoretical claims.

---

> ### Author Rebuttal · Authors · 2025-04-01
>
> Thank you for your valuable insights, which are fundamental to strengthening our paper. We handle all questions in their given order and will incorporate those details in the revision.
>
> Q1: more ablation studies on different training techniques.
>
> A1: We randomly sampled 1200 Video-Prompt pairs from WISA and perform lora(r=128) fine-tuning on CogVideoX 5B. The results shown below indicated that it does not solve the problem of model understanding of physical laws. This may be due to:
>
> - The training set is too small, not covering enough domains.
> - The base model's capabilities are insufficient, making it difficult to generalize.
> - More explicit injection of physical laws is needed, such as using synthetic videos.
>
> | Model | Mechanics | Optics | Thermal | Material | Avg |
> | --- | --- | --- | --- | --- | --- |
> | CogVideoX 5B | 0.39 | 0.55 | 0.40 | 0.42 | 0.449 |
> | CogVideoX 5B （FT） | 0.38 | 0.58 | 0.41 | 0.40 | 0.453 |
>
> Q2: its applicability to unseen, real-world scenarios remains unclear.
>
> A2: We first discussed the effectiveness of PhyGenEval on newly added prompts. Specifically, we extracted 50 prompts from WISA and applied 8 open-source models to generate videos. For each model, we extracted 20 videos and perform human evaluation. We calculated the human alignment coefficient(spearman) with machine scores in the table below, PhyGenEval has a certain generalization ability beyond PhyGenBench prompts.
>
> | Mechanics | Optics | Thermal | Avg |
> | --- | --- | --- | --- |
> | 0.71 | 0.76 | 0.76 | 0.74 |
>
> Next, we acknowledged that using real videos as references would have provided more reference signals, but this also presented several challenges: collecting real videos for each prompt was difficult, physical processes were diverse making definitive examples hard to find, and comparing videos with different frame rates required additional model training. Therefore, we adopted an alternative approach to incorporate real physical videos into PhyGenEval. Specifically, we extracted 50 Video-Caption pairs from WISA, belonging to mechanics, optics, and thermodynamics categories. We parsed the corresponding video captions into the prompt and question format used in PhyGenBench and evaluated them using PhysGenEval. The results showed that real videos achieved extremely high physical alignment scores under PhyGenEval, demonstrating the robustness of the framework.
>
> | Mechanics(17 samples) | Optics(17) | Thermal(16) |
> | --- | --- | --- |
> | 0.93 | 0.95 | 0.93 |
>
> We tested the performance of 8 models on these prompts, using machine scores and machine scores / real video machine scores, with the latter serving as a reference score to eliminate errors. The results were as follows. The spearman coefficient of model rankings calculated by these two methods is **0.90**, indicating that PhyGenEval can achieve robust results.
>
> |  | Mechanics | Optics | Thermal | Avg. |
> | --- | --- | --- | --- | --- |
> | CogVideoX2B | 0.36\0.37 | 0.44\0.49 | 0.33\0.41 | 0.39\0.39 |
> | CogVideoX5B | 0.36\0.36 | 0.52\0.57 | 0.47\0.53 | 0.46\0.50 |
> | OpensoraV1.2 | 0.36\0.38 | 0.49\0.52 | 0.39\0.39 | 0.43\0.45 |
> | Lavie | 0.23\0.27 | 0.42\0.45 | 0.36\0.40 | 0.36\0.38 |
> | Vchitect2.0 | 0.41\0.43 | 0.52\0.57 | 0.42\0.44 | 0.47\0.50 |
> | Hunyuan | 0.46\0.49 | 0.53\0.55 | 0.39\0.40 | 0.48\0.51 |
> | Pyramid Flow(flux) | 0.33\0.37 | 0.50\0.54 | 0.44\0.50 | 0.44\0.48 |
> | Pyramid Flow(sd3) | 0.43\0.54 | 0.46\0.52 | 0.33\0.40 | 0.42\0.49 |
>
> Q3: The evaluation method depends on CLIPScore, its accuracy is not well-verified.
>
> A3: First, we considered the possibility of inaccurate retrieval when designing our method. For instance, the calculation of $S_{key}$ at line 300, which includes $VLM(I_j,P_r)$, was specifically added as a regularization term to account for retrieval inaccuracies (as explained in line 304). Additionally, we pointed out that since we deliberately control the simplicity of scenes in PhyGenBench and require T2V models to semantically match the prompts, the success rate of CLIPScore retrieval aws relatively high. We provided an analysis of this in Appendix D.4 under "The robustness of retrieval operations."
>
> Q4: the framework does not explicitly assess motion coherence
>
> A4: We considered motion coherence to be a general video quality, but here we mainly focused on the correctness of physical laws. We tested the Temporal Quality - Motion Smoothness metric from VBench.  However, As shown in the table below, we found that the correlation coefficient between motion smoothness scores and human ratings of physical correctness is only 0.013, which made it challenging to use the metric as a distinguishing factor for supporting physics-based evaluation.
>
> | Min | Max | Avg | Spearman | Kendall |
> | --- | --- | --- | --- | --- |
> | 0.971 | 0.995 | 0.985 | 0.013 | 0.017 |
>
> We're truly grateful for your suggestions, and please let us know if any concerns arise.

---

> > ### Comment · Reviewer_QPju · 2025-04-06
> >
> > I appreciate the additional experiment the authors provided. Since the evaluated videos typically depict a single physical phenomenon with only a few co-occurring simple objects, I’m curious how the evaluation would hold up in more complex scenarios. Specifically, if multiple objects and different physical phenomena appear within the same video, would that impact the robustness of the evaluation? Additionally, how well would the CLIPScore capture the diversity across key frames in such cases?

---

> > > ### Author Response · Authors · 2025-04-07
> > >
> > > Thank you for your acknowledgment. We address all questions sequentially and will incorporate those details in the revisions.
> > >
> > > We recognize that evaluation in complex scenarios is more challenging, but complex physical scenes can sometimes be decomposed into several simple physics systems. Therefore, we started with the simplest physical scenarios to expose model limitations. As models evolve, we will explore more complex and diverse physical scenarios.
> > >
> > > We tested 8 open-source models using 50 prompts from WISA to generate videos and measured retrieval accuracy. The scenes in WISA differ from those in PhyGenBench and are mostly more complex. The retrieval accuracy results are shown in the table below. Although the overall retrieval success rate is lower than in PhyGenBench, the lowest still exceeds 0.5.
> > >
> > > | Model | Mechanics(17 samples) | Optics(17) | Thermal(16) |
> > > | --- | --- | --- | --- |
> > > | CogVideoX2B | 0.6006 | 0.6401 | 0.5772 |
> > > | CogVideoX5B | 0.5418 | 0.7963 | 0.6091 |
> > > | OpensoraV1.2 | 0.6472 | 0.7431 | 0.7056 |
> > > | Lavie | 0.5153 | 0.6550 | 0.5374 |
> > > | Vchitect2.0 | 0.6303 | 0.8006 | 0.6634 |
> > > | Hunyuan | 0.6832 | 0.8097 | 0.7253 |
> > > | Pyramid Flow(flux) | 0.6874 | 0.7974 | 0.6638 |
> > > | Pyramid Flow(sd3) | 0.6824 | 0.7476 | 0.6550 |
> > >
> > > In addition, we randomly selected 20 videos from the 50 videos generated by each model. Human annotators were asked to score these videos, focusing on physical correctness. We calculated the Spearman correlation between the PhyGenEval scores and the human scores. The results are shown in the table below:
> > >
> > > | Mechanics | Optics | Thermal | Avg |
> > > | --- | --- | --- | --- |
> > > | 0.71 | 0.76 | 0.76 | 0.74 |
> > >
> > > Even though the retrieval success rate has decreased, the machine results still maintain high similarity with human ratings. We believe this is due to our three-stage evaluation design framework and the regularization term for retrieval errors (The calculation of $S_{key}$ at line 300), which provides some correction for retrieval errors. This demonstrates that PhyGenEval remains robust for more complex scenes that are not part of PhyGenBench.
> > >
> > > With the development of models, we will explore more diverse and complex scenarios. For more complex scenarios, we consider the following approaches to enhance retrieval robustness:
> > >
> > > 1. Use stronger VLMs to enhance PhyGenEval, e.g., replacing CLIPScore with more powerful video VLMs such as InternVideo2.5.
> > >
> > > 2. Decouple complex physical phenomena into multiple simple physical phenomena. For example, in billiard ball collisions, analyze frame by frame whether the collisions between pairs of balls conform to physical laws such as conservation of momentum, implementing a multi-level evaluation framework.

---

### Decision · Program_Chairs · 2025-05-01

**Decision:**

Accept (poster)

**Comment:**

This paper proposes a new benchmark for video models as "world simulators", quantifying some recent claims that T2V models could be important for embodied AGI research. The paper proposes a hierarchical approach which tackles multiple dimensions of the problem and makes it clear as to which models work where. This work could become an important benchmark in a rapidly expanding field and lays the bedrock for more rigorous evaluation. All reviewers recommended accept and authors addressed feedback constructively.